# Multi-Modal Inverse Constrained Reinforcement Learning from a Mixture of Demonstrations

**Guanren Qiao**[1], **Guiliang Liu**[*1], **Pascal Poupart**[2,3], **Zhiqiang Xu**[4]

[1]School of Data Science, The Chinese University of Hong Kong, Shenzhen,
[2]University of Waterloo, [3]Vector Institute, [4]Mohamed bin Zayed University of Artificial Intelligence
`223040241@link.cuhk.edu.cn, liuguiliang@cuhk.edu.cn,`
`ppoupart@uwaterloo.ca,zhiqiang.xu@mbzuai.ac.ae`

## Abstract

Inverse Constraint Reinforcement Learning (ICRL) aims to recover the underlying constraints respected by expert agents in a data-driven manner. Existing ICRL algorithms typically assume that the demonstration data is generated by a single type of expert. However, in practice, demonstrations often comprise a mixture of trajectories collected from various expert agents respecting different constraints, making it challenging to explain expert behaviors with a unified constraint function. To tackle this issue, we propose a Multi-Modal Inverse Constrained Reinforcement Learning (MMICRL) algorithm for simultaneously estimating multiple constraints corresponding to different types of experts. MMICRL constructs a flow-based density estimator that enables unsupervised expert identification from demonstrations, so as to infer the agent-specific constraints. Following these constraints, MMICRL imitates expert policies with a novel multi-modal constrained policy optimization objective that minimizes the agent-conditioned policy entropy and maximizes the unconditioned one. To enhance robustness, we incorporate this objective into the contrastive learning framework. This approach enables imitation policies to capture the diversity of behaviors among expert agents. Extensive experiments in both discrete and continuous environments show that MMICRL outperforms other baselines in terms of constraint recovery and control performance. Our implementation is available at: https://github.com/qiaoguanren/Multi-Modal-Inverse-Constrained-Reinforcement-Learning.

## 1 Introduction

A fundamental prerequisite for achieving safe Reinforcement Learning (RL) is that the agents' policies must adhere to the underlying constraints in the environment [1, 2]. However, in many real-world applications (e.g., robot control and autonomous driving), the ideal constraints are time-varying, context-dependent, and inherent to experts' own experience. These constraints are hard to specify mathematically and may not be readily available to RL agents in policy updates.

A promising approach for learning latent constraints is Inverse Constraint Reinforcement Learning (ICRL) [3, 4, 5]. As a data-driven technique, ICRL recovers the underlying constraints

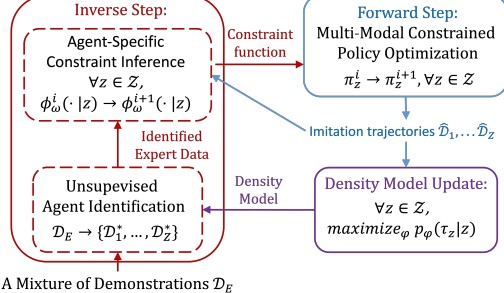

Figure 1: The flowchart of MMICRL.

---

*Corresponding author: Guiliang Liu, email: liuguiliang@cuhk.edu.cn

37th Conference on Neural Information Processing Systems (NeurIPS 2023).

respected by expert agents from their demonstrations and utilizes these constraints to support downstream applications. Existing ICRL methods [3, 4, 6, 7, 8] commonly assumed that all expert demonstrations follow the same constraints, and they approximated these constraints with a unified constraint model, whereas, in practice, the demonstration data may be collected from various agents, and these agents might follow different or even conflicting constraints. It is problematic to leverage a single constraint model to explain the behaviors of diverse agents. For example, in the context of autonomous driving [9], the vehicle-distance constraints followed by trucks and cars should differ, and misapplying these constraints could lead to serious traffic accidents.

To differentiate expert demonstrations, previous RL approaches [10, 11, 12, 13] inferred the latent structure of expert demonstrations and identified expert agents by examining their behavioral preferences. However, these methods were specifically designed for imitation learning rather than constraint inference. Moreover, as their identification process primarily relies on an agent classifier that evaluates state-action pairs, there is no theoretical guarantee that the optimal model is identifiable.

In this work, we propose the Multi-Modal Inverse Constrained Reinforcement Learning (MMICRL) algorithm for estimating agent-specific constraints from a mixture of expert demonstrations (Figure 1). MMICRL extends the traditional maximum entropy framework [14] with agent-conditioned entropy minimization and unconditioned entropy maximization subject to a permissibility constraint. The resulting policy representation facilitates inferring agent-specific constraints by alternating between the following steps: 1) *Unsupervised Agent Identification*. MMICRL conducts trajectory-based agent identification using a flow-based density estimator. The optimal results are identifiable since each agent's policy must correspond to a unique occupancy measure [15]. 2) *Agent-Specific Constraint Inference*. Leveraging the identified demonstrations, MMICRL estimates a permissibility function for distinguishing expert demonstrations from sub-optimal ones, based on which we construct constraints for each type of agent. 3) *Multi-Modal Policy Optimization*. MMICRL measures the accuracy of inferred constraint models by comparing the similarity between expert trajectories and those generated by imitation policies under the inferred constraints. To capture the diversity of behaviors exhibited by multiple agents, we incorporate policy optimization within the contrastive learning framework. We treat the generated trajectories as noisy embeddings of agents, which serve as compact representations of their behaviors. Utilizing the contrastive estimation methods [16], we can enhance the similarity between embeddings for agents of the same type, while simultaneously maintaining the distinctiveness of diverse agent types.

We empirically demonstrate the performance of our method by conducting experiments in both discrete (e.g., Gridworld) [7] and continuous environments (e.g., MuJoCo) [17]. MMICRL significantly surpasses other baselines in terms of distinguishing different agents, adhering to the true constraints, and optimizing control performance. To examine the robustness of MMICRL , we investigate its ability to recover from incorrect agent identification and subsequently infer the correct constraints.

## 2  Related Works

**Inverse Constrained Reinforcement Learning.** Prior ICRL methods typically learned constraints from demonstrations under the maximum entropy framework [14]. Some research [18, 19] employed constraints to differentiate between feasible and infeasible state-action pairs in Constrained MDP, but these studies were restricted to inferring discrete constraints in environments with known dynamics. A subsequent work [4] extended this approach to continuous state-action spaces with unknown transition models by utilizing neural networks to approximate constraints. Inspired by Bayesian Inverse Reinforcement Learning [20, 21, 22], [8] inferred probability distributions over constraints. To better model demonstrations, [6] extended ICRL to infer soft constraints rather than hard ones, and [23] explored ICRL under the multi-agent setting. Striving for efficient comparisons, [7] established an ICRL benchmark across various RL domains, such as Gridworld, robot control, and autonomous driving. However, these algorithms primarily target inferring constraints for a single agent type, without considering the distinct constraints associated with multiple agent types.

**Learning from a Mixture of Expert Demonstrations.** Multi-task Inverse Reinforcement Learning (IRL) [24, 25] aims to learn from a mixture of expert demonstrations. Some previous studies [10, 26, 12, 27] utilized the Generative Adversarial Imitation Learning algorithm [28] to model the behaviors of multiple agents. [10, 11, 13, 29] learned interpretable representations of behavioral policies and inferred the latent representation of expert demonstrations in an unsupervised way. Specifically, [30] learned a Variational Auto-Encoder (VAE) [31] where the encoder infers the latent

factors of variation from mixed demonstrations and the decoder models different types of expert behaviors. To strengthen the consistency between the learned policy and the types of agents, [12, 32] included "burn-in demonstrations" for updating the imitation policies. These methods, however, were proposed for imitation learning or rewards recovery (i.e., for IRL) [33] instead of constraint inference.

## 3 Problem Definition

**Constrained Mixture-Agent Markov Decision Process.** To support constraint inference for different agents, we formulate the environment as a Constrained Mixture-Agent Markov Decision Process (CMA-MDP) [34] $\mathcal{M}^\phi$, which can be defined by a tuple $(\mathcal{S}, \mathcal{A}, \mathcal{Z}, \mathrm{R}, p_\mathcal{T}, \{(p_{\mathcal{C}_i}, \epsilon_i)\}_{\forall i}, \gamma, \mu_0)$ where: 1) $\mathcal{S}$, $\mathcal{A}$ and $\mathcal{Z}$ denote the space of states, actions, and latent code (for specifying expert agents). 2) $p_\mathcal{T}(s'|s,a)$ and $\mathrm{R}(s,a)$ define the transition and reward functions. 3) $p_{\mathcal{C}_j}(c|s,a,z)$ refers to an agent-specific cost model with an associated bound $\epsilon_j(z)$, where $j$ indicates the index of a constraint. 4) $\gamma$ is the discount factor and $\mu_0(s)$ defines the initial state distribution. CMA-MDP assumes the agents are differentiable by examining their policies, implying that different agent types cannot share identical policies. We aim to elucidate these differences with the constraints associated with each agent type.

In contrast to Multi-Agent Reinforcement Learning (MARL) [35], where multiple agents can act concurrently, CMA-MDP allows only one agent to operate at a given time. Nevertheless, since the agents in CMA-MDP might adhere to distinct constraints, they can develop different optimal control policies or strategies. However, standard MARL frameworks do not explicitly differentiate between agent types, nor do they distinguish their respective policies and constraints.

**Policy Update under Conditional Constraints.** We introduce Constrained Reinforcement Learning (CRL) based on CMA-MDP. For an agent identified by $z$, the goal of CRL is to find a policy $\pi^*(a|s,z)$ that maximizes expected discounted rewards under the conditional constraints:

$$\mathcal{J}(\pi|z) = \max_\pi \mathbb{E}_{\mu_0, p_\mathcal{T}, \pi} \Big[ \sum_{t=0}^T \gamma^t r_t \Big] + \beta \mathcal{H}(\pi) \text{ s.t. } \mathbb{E}_{\tau \sim (\mu_0, p_\mathcal{T}, \pi), p_{\mathcal{C}_j}} \Big[ c_j(\tau|z) \Big] \le \epsilon_j(z) \ \forall j \in [0, J] \quad (1)$$

where $\mathcal{H}(\pi)$ denotes the policy entropy weighted by $\beta$, and we follow [4] to define the trajectory cost $c(\tau|z) = 1 - \prod_{(s,a) \in \tau} \phi(s,a|z)$ where the permissibility function $\phi(s,a|z)$ indicates the probability that performing action $a$ under a state $s$ is safe for agent $z$. CRL commonly assumes that the constraints are known, whereas in practice, instead of directly observing the constraint signals, we often have access to expert demonstrations that follow the underlying constraints, and thus the agent must recover the constraints from the demonstration dataset.

**Constraint Inference from a Mixture of Expert Dataset.** Based on the CMA-MDP, the goal of ICRL is to discover the underlying constraints respected by different types of expert agents from their demonstrations. To achieve it, we consider the Maximum Entropy framework [4, 14] and represent the likelihood function as follows:

$$p(\mathcal{D}_E|\phi) = \frac{1}{(Z_{\mathcal{M}^{\hat{c}_\phi}})^N} \prod_{i=1}^N \sum_z \mathbb{1}_{\tau^i \in \mathcal{D}_z} \exp\Big[ r(\tau^{(i)}) \Big] \mathbb{1}^{\mathcal{M}^{\hat{c}_\phi}}(\tau^{(i)}|z) \quad (2)$$

where 1) $N$ denotes the number of trajectories in the demonstration dataset, 2) $Z_{\mathcal{M}^{\hat{c}_\phi}}$ is a normalizing term, 3) the permissibility indicator $\mathbb{1}^{\mathcal{M}^{\hat{c}_\phi}}(\tau^{(i)})$ can be defined by $\phi(\tau^{(i)}|z) = \prod_{t=1}^T \phi_t(s_t^i, a_t^i|z)$, and 4) the agent identifier $\mathbb{1}_{\tau^{(i)} \in \mathcal{D}_z}$ determines whether the trajectory $\tau^{(i)}$ is generated by the agent $z$.

By following [4], constraint inference can be formulated as inferring $\phi_t$ by maximizing this likelihood function. A common approach is parameterizing $\phi_t$ with neural network models and updating the model parameters with the agent-specific expert data [4, 6, 7]. However, the major challenge of this task lies in our lack of knowledge about the agent's identity that generates the expert trajectories (i.e., $\mathbb{1}_{\tau^{(i)} \in \mathcal{D}_z}$ is unknown). For example, a vehicle trajectory dataset does not label the specific type of vehicles, or these labels are coarse-grained, incomplete, and noisy. On the other hand, to explain the optimal policy with the learned constraints, *the types of agents and the constraints must be consistent*. If we evaluate the policy of an agent $z$ under the constraint for another type of agent $z'$ (e.g., evaluate the driving strategy of a truck under the constraint for a car), the performance will be substantially compromised. As a result, for a specific agent $z$, the expert trajectories $\{\tau_z^*\}$ are optimal while the trajectories generated by other experts $\{\tau_{z'}^*, \tau_{z''}^*, \tau_{z'''}^*, ...\}$ are supposed to be sub-optimal

or even infeasible. To capture the correct constraints, the ICLR algorithm must identify the agent corresponding to each trajectory in an unsupervised manner.

## 4 Inverse Constrained Reinforcement Learning for a Mixture of Experts

In order to infer constraints for various expert agent types, we introduce the Multi-Modal Inverse Constrained Reinforcement Learning (MMICRL) algorithm (Figure 1). MMICRL employs a conditional imitation policy to capture the diverse agent behaviors, facilitating unsupervised agent identification. Technically, we require the agents' policies to 1) exhibit high entropy when the agent type (specified by $z$) is unknown, and 2) collapse to a specific behavioral mode when the type is determined. The objective for MMICRL can be expressed as :

$$\text{Minimize} - \alpha_1 \mathcal{H}[\pi(\tau)] + \alpha_2 \mathcal{H}[\pi(\tau|z)] \tag{3}$$

$$\text{Subject to} \int \pi(\tau|z) f_z(\tau) \mathrm{d}\tau = \frac{1}{N} \sum_{\tau \in \mathcal{D}_z} f(\tau), \ \int \pi(\tau|z) \mathrm{d}\tau = 1, \text{ and } \int \pi(\tau|z) \log \phi(\tau|z) \mathrm{d}\tau \geq \epsilon$$

(Proof is provided in Appendix B.) where $f(\cdot)$ represents a latent feature extractor, $\mathcal{H}[\pi(\tau|z)]$ denotes the agent-specific policy entropy, and $\mathcal{H}[\pi(\tau)]$ signifies the entropy without knowledge of the agent type. The weighting parameters $\alpha_1$ and $\alpha_2$ (both $\geq 0$) determine the balance between conditional entropy minimization and general entropy maximization. This objective differs from the traditional Maximum Entropy Inverse Reinforcement Learning (MEntIRL) [14] in two ways: 1) the objective also minimizes an entropy conditioned on agent types, and 2) it incorporates an additional constraint related to the policy's permissibility (the last constraint). Given this objective, the optimal representation for the trajectory likelihood (i.e., trajectory policy) is:

**Proposition 4.1.** *Let $p(z|\tau)$ denote the trajectory-level agent identifier, let $r(\tau) = \frac{\lambda_0}{\alpha_2 - \alpha_1} f(\tau)$ denote the trajectory rewards, let $Z_{\mathcal{M}^{\hat{c}_\phi}}$ denote a normalizing term. The optimal policy of the above optimization problem can be represented as:*

$$\pi(\tau|z) = \frac{1}{Z_{\mathcal{M}^{\hat{c}_\phi}}} \exp\left[\frac{\alpha_1 \mathbb{E}_{z \sim p(z)}[\log(p(z|\tau))]}{\alpha_2 - \alpha_1} + r(\tau)\right] \phi(\tau|z)^{\frac{\lambda_2}{\alpha_1 - \alpha_2}} \tag{4}$$

Building upon this policy representation (4), we present the key steps of MMICRL , which involves iteratively performing: 1) unsupervised agent identification for calculating $p(z|\tau)$ (Section 4.1), 2) conditional inverse constraint inference (Section 4.2) for deducing $\phi(\tau|z)$, and 3) multi-modal policy update (Section 4.3) for approximating $\pi(\tau|z)$. MMICRL alternates between these steps until the imitation policies reproduce expert trajectories, signifying that the inferred constraints align with the ground-truth constraints.

### 4.1 Unsupervised Agent Identification

MMICRL identifies expert trajectories (i.e., learning $p(z|\tau)$) in an unsupervised manner. Previous works commonly determined the agents' identities by examining the state-action features [10, 11, 12] with the classifier $p(z|s_z, a_z)$. Nevertheless, different agents may exhibit similar behaviors under certain contexts or at specific time steps within a trajectory, which makes this point-wise identification problematic (e.g., see our experimental results of InfoGAIL [10] in Section 5).

To derive a reliable agent identifier, MMICRL performs trajectory-level identification by estimating an agent-specific density. Specifically, we define a state-action density (i.e., normalized occupancy measure) $\rho_\pi(s, a) = (1 - \gamma)\pi(a|s) \sum_{t=0}^{\infty} \gamma^t p(s_t = s|\pi)$ where $p(s_t = s|\pi)$ is the probability density of state $s$ at time step $t$ following policy $\pi$. Based on this density, we consider the theorem:

**Proposition 4.2** (Theorem 2 of [15]). *Suppose $\rho$ is the occupancy measure that satisfies the Bellman flow constraints: $\sum_a \rho(s, a) = \mu_0(s) + \gamma \sum_{s', a} \rho(s', a) P\tau(s'|s, a)$ and $\rho(s, a) \geq 0$. Let the policy defined by: $\pi_\rho(a|s) = \frac{\rho(s, a)}{\int \rho(s, a') \mathrm{d}a'}$, then $\pi_\rho$ is the only policy whose occupancy measure is $\rho$.*

**Density Estimation.** The aforementioned theorem provides a crucial insight for agent identification: "*one can identify an expert agent by examining the occupancy measures in the expert trajectories*". Leveraging this insight, we design a state-action density estimator to compute a density using a *Conditional Flow-based Density Estimator (CFDE)*. CFDE estimates the density of input variables in the training data distribution under an auto-regressive constraint. Moreover, to enhance our

density estimator's sensitivity to the behavior of different agents, the estimator also conditions on the agent type. so $p(\boldsymbol{x}|z) = \prod_i p(\boldsymbol{x}_i|\boldsymbol{x}_{1:i-1}, z)$ where $\boldsymbol{x} := (s, a)$ defines an event. We implement $p(\boldsymbol{x}_i|\boldsymbol{x}_{1:i-1}, z) = \mathcal{N}(\boldsymbol{x}_i|\mu_i, (\exp(\alpha_i))^2)$ where $\mu_i = \psi_{\mu_i}(\boldsymbol{x}_{1:i-1}, z)$ and $\alpha_i = \psi_{\alpha_i}(\boldsymbol{x}_{1:i-1}, z)$. The neural function $\psi$ is implemented by stacking multiple MADE layers [36]. The corresponding agent identifier can be represented by Bayesian rule, so $p_\psi(z|\tau) = \frac{p(z) \cdot p_\psi(\tau|z)}{\sum_z p(z) \cdot p_\psi(\tau|z)}$. We assume a uniform prior $p(z)$, thereby deriving the following form: $p_\psi(z|\tau) = \frac{\prod_{(s,a) \in \tau} p_\psi(s,a|z)}{\sum_{z'} \prod_{(s,a) \in \tau} p_\psi(s,a|z')}$.

**Agent Identification.** After learning the density model $p_\psi(s, a|z)$ with CFDE, we divide $\mathcal{D}_E$ into sub-datasets $\{\mathcal{D}_z\}_{z=1}^{|\mathcal{Z}|}$ by: 1) initializing the dataset $\mathcal{D}_z = \emptyset$ and 2) $\forall \tau_i \in \mathcal{D}_E$, adding $\tau^i$ into $\mathcal{D}_z$ if $z = \arg\max_z \sum_{(s,a) \in \tau_i} \log[p_\psi(s,a|z)]$. We repeat the above steps for all $z \in \mathcal{Z}$.

### 4.2 Agent-Specific Constraint Inference

Based on the identified expert dataset, we have $p_\omega(z|\tau_z) = 1, \forall \tau_z \in \mathcal{D}_z$, and thus $\log p_\omega(z|\tau_z) = 0$, so the likelihood function (4) can be simplified to:

$$p(\mathcal{D}_z|\phi, z) = \prod_{i=1}^{N} \frac{1}{Z_{\mathcal{M}^{\hat{c}_\phi}}} \exp\left[r(\tau_z^{(i)})\right] \phi(\tau_z^{(i)})^{\frac{\lambda_2}{\alpha_1 - \alpha_2}} \tag{5}$$

where $\phi(\tau_z^{(i)}) = \prod_{(s,a) \in \tau_z^{(i)}} \phi_t(s,a|z)$ and the normalizer $Z_{\mathcal{M}^{\hat{c}_\phi}} = \int \exp[r(\tau)]\phi(\tau)^{\frac{\lambda_2}{\alpha_1 - \alpha_2}} d\tau$. By defining $\eta = \frac{\lambda_2}{\alpha_1 - \alpha_2}$, we can then define the log-likelihood $\log[p(\mathcal{D}_z|\phi, z)]$ as:

$$\sum_{i=1}^{N}\left[r(\tau_z^{(i)}) + \eta \log \prod_{t=0}^{T} \phi(s_t^{(i)}, a_t^{(i)}|z)\right] - N \log \int \exp[r(\hat{\tau})]\left[\prod_{t=0}^{T} \phi(\hat{s}_t, \hat{a}_t|z)\right]^\eta d\hat{\tau} \tag{6}$$

We parameterize the instantaneous permissibility function with $\omega$, i.e., construct $\phi_\omega(s_t, a_t|z)$ and update the parameters by computing the gradient of the above likelihood function (the derivation resembles that of [4]), so $\nabla_\omega \log[p(\mathcal{D}_z|\phi, z)]$ can be defined as:

$$\sum_{i=1}^{N}\left[\nabla_\phi \sum_{t=0}^{T} \eta \log[\phi_\omega(s_t^{(i)}, a_t^{(i)}|z)]\right] - N\mathbb{E}_{\hat{\tau} \sim \pi_{\mathcal{M}^\phi}(\cdot|z)}\left[\nabla_\phi \sum_{t=0}^{T} \eta \log[\phi_\omega(\hat{s}_t, \hat{a}_t|z)]\right] \tag{7}$$

This inverse constraint objective relies on the nominal trajectories $\hat{\tau}$ sampled with the conditional policy $\pi_{\mathcal{M}^{\hat{\phi}}}(\tau|z)$ (also see Figure 1). For simplicity, we denote it as $\pi(\tau|z)$. In the following, we will introduce our approach to learning $\pi(\tau|z)$.

### 4.3 Multi-Modal Policy Optimization

By definition, the policy $\hat{\pi}(\tau|z)$ is trained to maximize cumulative rewards subject to constraint $\mathbb{E}_{\tau \sim \pi(\cdot|z)}[\log \phi(\tau|z)] \geq \epsilon$. To be consistent with our MMICRL objective in formula (3), we design the multi-modal policy optimization objective in the following:

$$\tag{8}$$
$$\min_\pi -\mathbb{E}_{\pi(\cdot|z)}\left[\sum_{t=0}^{T} \gamma^t r(s_t, a_t)\right] - \alpha_1 \mathcal{H}[\pi(\tau)] + \alpha_2 \mathcal{H}[\pi(\tau|z)] \text{ s.t. } \mathbb{E}_{\pi(\cdot|z)}\left(\sum_{t=0}^{h} \gamma^t \log \phi_\omega(s, a, z)\right) \geq \epsilon$$

This objective extends maximum entropy policy optimization by minimizing an additional agent-conditioned entropy $\mathcal{H}[\pi(\tau|z)]$, which limits the variance of policy distribution for a specific type of agent. The balance between these entropy terms is controlled by $\alpha_1$ and $\alpha_2$. Since $\mathcal{H}[\pi(\tau)] = \mathcal{H}[\pi(\tau|z)] + \mathbb{E}_{z \sim p(z), \tau \sim \pi(\tau|z)}\left[\log[p_\psi(z|\tau)]\right] + \mathcal{H}(z)$, and by removing the uniform prior $p(z)$ (since it is independent of the policy update), the objective (8) can be simplified to:

$$\tag{9}$$
$$\min_\pi -\mathbb{E}_{\pi(\cdot|z)}\left[r(\tau) + \alpha_1 \log[p_\psi(z|\tau)]\right] + (\alpha_2 - \alpha_1)\mathcal{H}[\pi(\tau|z)] \text{ s.t. } \mathbb{E}_{\pi(\cdot|z)}\left(\sum_{t=0}^{h} \gamma^t \log \phi_\omega(s, a, z)\right) \geq \epsilon$$

Intuitively, this objective expands the reward signals with a log-probability term $\log[p_\psi(z|\tau)]$, which encourages the policy to generate trajectories from high-density regions for a specific agent type. This approach ensures that the learned policies $\pi(\cdot|z)_{z=1}^{Z}$ are differentiable.

**Learning Diverse Policies via Contrastive Estimation.** In practice, directly augmenting the reward with a log-probability term (as in objective 9) may lead to a sub-optimal policy [7]. This issue

arises because the log-probability term assigns a large penalty to trajectories with low $p_\psi(z|\tau)$. In such cases, the controlling policy becomes more sensitive to density estimation (since $p_\psi(z|\tau) = \text{softmax}[p_\psi(\tau|z)]$, see Section 4.1) rather than the reward signals. Balancing the trade-off between reward and density maximization by identifying an optimal weight $\alpha_1$ is challenging, especially when predictions from $p_\psi(\tau|z)$ are less accurate at the beginning of training (i.e., during the cold start).

To resolve the above issues, we consider replacing the identification probability with a contrastive estimation method by constructing the following objective for policy optimization:

$$\min_\pi -\mathbb{E}_{\pi(\cdot|z)}\Big(r(\tau) + \alpha_1 L_{ce}(\tau, \mathcal{V}_{1,\ldots,|Z|})\Big) + (\alpha_2 - \alpha_1)\mathcal{H}(\pi(\tau|z)) \text{ s.t. } \mathbb{E}_{\pi(\cdot|z)}\Big(\sum_{t=0}^h \gamma^t \log \phi_\omega(s,a,z)\Big) \geq \epsilon \quad (10)$$

where $\mathcal{V}$ defines the *probing sets* (constructed with the density estimator in Algorithm 1). Given a specific agent type $z$, these probing vectors are among the most representative data points since they are located in a high-density region conditioning on $z$ and a low-density region conditioning on other agent types ($\tilde{z} \neq z$). Inspired by the InfoNCE loss [16, 37] , $L_{ce}$ can be calculated as:

$$L_{ce}(\tau, \mathcal{V}_{1,\ldots,|Z|}) = \sum_{t=0}^T \log \frac{\exp\Big[\sum_{(\hat{s}_z, \hat{a}_z) \in \mathcal{V}_z} f_s[(s_t, a_t), (\hat{s}_z, \hat{a}_z)]\Big]}{\sum_{\tilde{z} \in \mathcal{Z}} \exp\Big[\sum_{(\tilde{s}, \tilde{a}) \in \mathcal{V}_{\tilde{z}}} f_s[(s_t, a_t), (\tilde{s}, \tilde{a})]\Big]} \quad (11)$$

where $f_s$ denotes the score function for measuring the similarity between the features from different state-action pairs (we use cosine similarity in the experiment). To interpret $L_{ce}(\tau, \mathcal{V}_{1,\ldots,|Z|})$, we can treat $(s,a) \in \{\tau, \mathcal{V}_z\}$ as positive embeddings for $\pi(\cdot|z)$ since they are generated by this policy. On the other hand, $(\tilde{s}, \tilde{a}) \in \{\mathcal{V}_{\tilde{z}}\}_{\tilde{z} \neq z}$ are negative embeddings for $\pi(\cdot|z)$ since they are generated by controlling with other policies $\pi(\cdot|\tilde{z})$ (where $\tilde{z} \neq z$). Considering the generation is influenced by the stochasticity in environment dynamics (e.g., transition functions), we can *equivalently view the generation process as injecting the environmental noise into the policy*, and thus Noise Contrastive Estimation (NCE) [38] becomes a compatible tool for learning differentiable policies. Specifically, since embeddings in $\mathcal{V}_z$ belong to a high conditional density region in $p(\cdot|z)$ (see Algorithm 1), the knowledge from the density estimator has been integrated into policy updates.

In essence, the integration of contrastive learning into policy optimization helps the algorithm to better understand the relationships between agents, their behaviors, and the corresponding expert trajectories, resulting in improved performance for tasks involving diverse agent types. *Algorithm 2 introduces the detailed implementation of MMICRL* .

---

**Algorithm 1:** Probing_Sets

---

**Input:** Agent type $z$, trajectory dataset $\mathcal{D}_z$ and conditional density model $p(\cdot|z)$
Initialize a probing sets $\mathcal{V}_z = \{\emptyset\}$;

**for** $\tau_z \in \hat{\mathcal{D}}_z$ **do**
$\quad$ Find $(\hat{s}, \hat{a}) = \arg\max_{(s,a) \in \tau_z} \log p_{\psi_i}(s,a|z) - \frac{1}{|\mathcal{Z}|-1}\sum_{\tilde{z}} \mathbb{1}_{\tilde{z} \neq z} \log p_{\psi_i}(s,a|\tilde{z})$;
$\quad$ Store the probing points $\mathcal{V}_z = \mathcal{V}_z \cup \{(\hat{s}, \hat{a})\}$;
**end**
**Output:** $\mathcal{V}_z$

---

## 5  Experiments

**Running Setting.** For a comprehensive comparison, we employ consistent evaluation metrics across all environments. These metrics include 1) *Constraint Violation Rate*, which assesses the likelihood of a policy violating a constraint in a given trajectory, and 2) *Feasible Cumulative Rewards*, which calculates the total rewards accumulated by the agent before violating any constraints. The demonstration data are assumed to have a zero violation rate. We run experiments with three different seeds and present the mean $\pm$ std results for each algorithm. To ensure a fair comparison, we maintain uniform settings for all comparison baselines. Appendix A.2 reports the detailed settings.

**Comparison Methods.** We employ an ablation strategy to progressively remove components from **MMICRL** (Algorithm 2) and consider the following variants: 1) **MMICRL-LD** excludes our contrastive estimation approach, using the objective (9) directly for policy updates. 2) Inspired by [10], **InfoGAIL-ICRL** replaces the trajectory density model with a discriminator to identify agents based on state-action pairs, extending GAIL [28] to distinguish between distinct agents' trajectories. 3) **MEICRL** [4] eliminates the agent identifier and expands traditional Maximum

**Algorithm 2:** Multi-Modal Inverse Constrained Reinforcement Learning (MMICRL)

---

**Input:** Expert demonstration $\mathcal{D}_E$, upper level iterations $N$, lower level iterations $B$
Initialize the policy $\pi_{\theta_0}(\cdot|\cdot, z)$, constraint function $\phi_{\omega_0}(\cdot|z)$ and density model $p_{\psi_0}(\cdot|z)$;
**for** $i = 1, 2, \ldots, N$ **do**
    Initialize an ensemble of imitation datasets $\{\hat{\mathcal{D}}_z\}_{z \in \mathcal{Z}}$ and each dataset $\hat{\mathcal{D}}_z = \{\emptyset\}$;
    **for** $j = 1, 2, \ldots, B$ **do**
        Randomly sample a $z_j$, and generate imitation trajectories $\tau_{z_j}$ with $\pi_{\theta_i}(\cdot|\cdot, z_j)$;
        Update the density model by maximizing: $\log p_{\psi_i}(\tau_{z_j}|z_j)$ ;
        Store the generated trajectories and the code $\hat{\mathcal{D}}_z = \hat{\mathcal{D}}_z \cup \{(\tau_{z_j}, z_j)\}$;
    **end**
    Divide $\mathcal{D}_E$ into sub-datasets $\{\mathcal{D}_{E,z}\}_{z=1}^{|\mathcal{Z}|}$ by utilizing the density model (see Section 4.1);
    **for** $k = 1, 2, \ldots, B$ **do**
        Sample an agent type from the prior $\hat{z} \sim p(z)$;
        Sample an expert trajectory $\tau_{E,\hat{z}}$ from the $\hat{z}^{th}$ expert dataset: $\tau_{E,\hat{z}} \sim \mathcal{D}_{E,\hat{z}}$;
        Sample a nominal trajectory from the $\hat{z}^{th}$ generated dataset: $\hat{\tau} \sim \hat{\mathcal{D}}_{\hat{z}}$;
        Update the constraint function $\phi_{\omega_i}$ with the objective (7);
    **end**
    **for** $z \in \mathcal{Z}$ **do**
        Construct a probing set $\mathcal{V}_z =$ProbingSets$(z, \mathcal{D}_z, p_\psi(\cdot|z))$(Algorithm 1);
        update the policy with the objective (10);
    **end**
**end**

---

Entropy (ME) IRL methods to infer Markovian constraints in a model-free environment. 4) **Binary Classifier Constraint Learning (B2CL)** constructs a deterministic binary classifier directly for constraint inference, bypassing the need for the maximum entropy framework.

## 5.1 Empirical Evaluations in Discrete Environments

The discrete environments we use are based on Gridworld, a widely studied RL environment that enables us to visualize the recovered constraints and trajectories generated by various agents. We create a 7x7 Gridworld map and design four distinct constraint map settings. In Figure 2, the leftmost column illustrates expert trajectories and constraints for each configuration. Notably, the first three settings (rows 1-3 in Figure 2) incorporate two different types of constraints, while the final setting (the last row in Figure 2) includes three constraint types. The primary goal for each agent is to navigate from the starting point to the endpoint while minimizing the number of steps and avoiding their respective constraints. To facilitate constraint inference, a demonstration dataset containing expert trajectories is provided for each environment [7].

Figure 2 displays the ground-truth trajectory map alongside the learned trajectory maps. Appendix C.1 summarizes the detailed performance in terms of feasible cumulative rewards, constraint violation rate, and destination-reaching rate. Without implementing agent identification, both B2CL and MEICRL can only restore a single constraint type. Although InfoGAIL-ICRL incorporates agent identification, its performance is unsatisfactory, resulting in incomplete destination-reaching. In contrast, MMICRL-LD and MMICRL exhibit significant improvements; they can identify various agent types corresponding to different constraints and generate accurate trajectories that successfully reach the endpoint. Notably, the enhancements provided by MMICRL are more prominent, and the trajectories produced by the MMICRL algorithm closely resemble expert demonstrations.

## 5.2 Empirical Evaluations in Continuous Environments

In continuous environments, we evaluate ICLR algorithms by whether they can infer location constraints for various types of robots (robots can only operate within designated areas). The task involves directing the robot to navigate out of the danger zone, represented by the constraint.

Table 1: Continuous environment setting.

| Setting | Dim. | $1^{st}$ constraint | $2^{nd}$ constraint |
|---|---|---|---|
| Half-cheetah | 24 | x$\leq -3$ | x$\geq 3$ |
| Antwall | 121 | x$\leq -3$ | x$\geq 3$ |
| Swimmer | 12 | x$\leq -0.01$ | x$\geq 0.01$ |
| Walker | 24 | x$\leq -0.1$ | x$\geq 0.1$ |

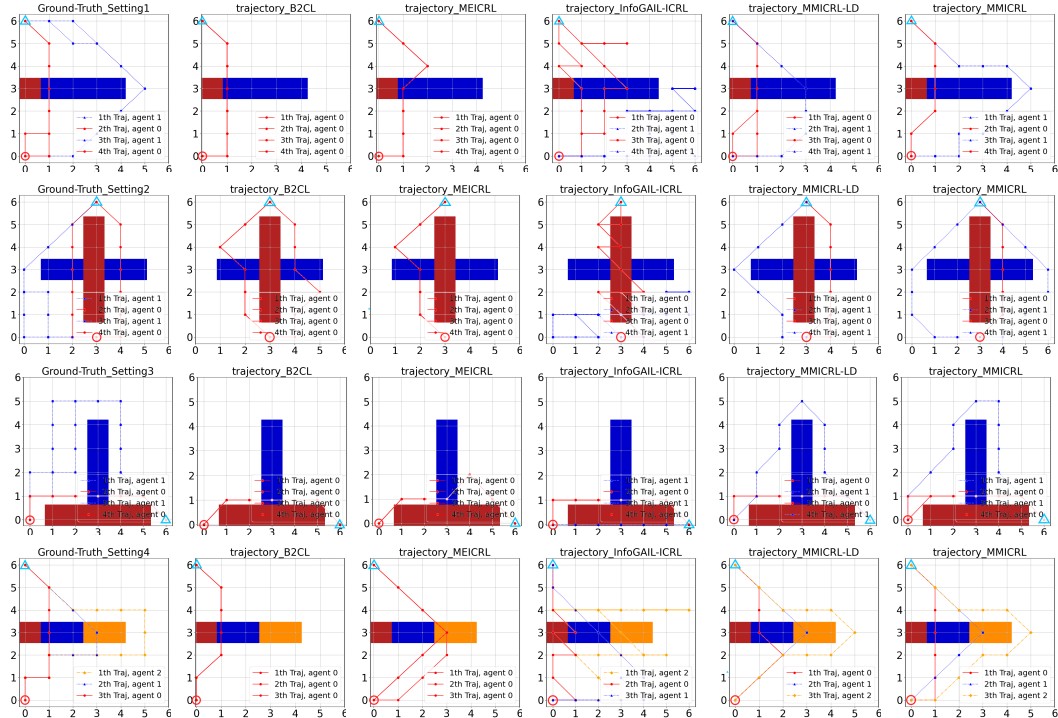

Figure 2: We utilize blue, red, and orange colors to represent the constraints or trajectories corresponding to each agent type. The red circle and blue triangle indicate the starting and ending points of each environment, respectively. Since MECL and B2CL recover only a single, unified constraint, they produce just one type of trajectory (Marked by the red color).

We base our continuous environments on MuJoCo [17]. For each environment, we configure two types of robots—one moving forward and the other moving backward—each with opposing constraints. Table 1 provides an overview of the settings. Our goal is to distinguish their respective constraints and enable each robot type to maximize rewards without violating constraints.

Figure 4 and Table 2 illustrate the constraint violation rate and feasible cumulative rewards for each agent across various environments. To evaluate the accuracy of the discovered constraints, Figure 3 illustrates the distribution of x-coordinate values for all states visited by agents of different types. Compared to other baselines, MMICRL consistently exhibits lower constraint violation rates and higher cumulative rewards, averaged over all agents. This superiority can be attributed to MMICRL's ability to capture the diversity of constraints by inferring agent-specific constraints and learning heterogeneous policies through contrastive estimation. While certain baselines may achieve higher cumulative rewards for a specific agent, none of these

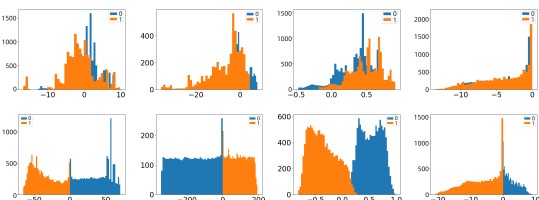

Figure 3: From left to right, the environments are Blocked Antwall, Blocked Half-Cheetah, Blocked Swimmer, and Blocked Walker. The first row shows the results of InfoGAIL-ICRL, followed by MMICRL. The Blue and orange areas represent the trajectories of two types of agents.

algorithms consistently outperform the others. (refer to Appendix C.2 for other complete results)

## 5.3 Robust Test: Can MMICRL Recover from the Early Mistakes?

To test the robustness of MMICRL, we enforce MMICRL to make incorrect agent identifications and assess whether MMICRL can recover from these errors. Specifically, in the first round of training, rather than using our density estimator, we randomly assign incorrect labels to 20% of expert trajectories. These incorrect identifications can impact ongoing constraint inference and policy updates. To recover from these mistakes, the MMICRL algorithm must accurately model diverse agent preferences and rectify the incorrect trajectories to match the appropriate agent type.

Table 2: MuJoCo testing performance. We report the average feasible rewards and constraint violation rate in 100 runs. The best average performance is highlighted in bold.

| Method | Blocked Half-Cheetah | Blocked Ant | Blocked Swimmer | Blocked Walker |
|---|---|---|---|---|
| **Feasible Cumulative Rewards** | | | | |
| B2CL | 5.03E+1 / 2.12E+3 | 1.33E+4 / 3.11E+3 | 3.28E+2 / 7.40 E-1 | 2.19E+1 / 7.88E+1 |
| MEICRL | 4.85E+1 / 3.45E+3 | 1.80E+4 / 3.31E+3 | 2.01E+2 / 8.30E-1 | 2.61E+1 / 7.33E+1 |
| InfoGAIL-ICRL | 2.22E+2 / 1.33E+2 | 3.73E+2 / 1.11E+2 | 2.38E+1 / 7.90E-1 | 2.15E+1 / 1.58E+3 |
| MMICRL-LD | 4.32E+3 / 2.56E+3 | 1.82E+4/ 2.21E+4 | 2.66E+2 / 6.10E+2 | 9.02E+2 / 5.12E+2 |
| MMICRL | **6.12E+3 / 3.06E+3** | **2.13E+4 / 2.17E+4** | **4.07E+2 / 6.48E+2** | **8.98E+2 / 1.53E+3** |
| **Constraint Violation Rate** | | | | |
| B2CL | 100%±0% / 67%±24% | 33%±24% / 67±24% | 62%±12% / 100%±0% | 100%±0% / 100%±0% |
| MEICRL | 100%±0% / 50%±25% | 0%±0% / 67±24% | 79%±14% / 100%±0% | 100%±0% / 100%±0% |
| InfoGAIL-ICRL | 25%±16% / 93%±5% | 34%±18% / 37%±19% | 73%±12% / 100%±0% | 100%±0% / 0%±0% |
| MMICRL-LD | 33%±24% / 34%±24% | **0%±0% / 0%±0%** | 71%±16% / 34%±23% | 52%±25% / 50%±25% |
| MMICRL | **0%±0% / 0%±0%** | **0%±0% / 0%±0%** | **55%±23% / 28%±19%** | **31%±22% / 25%±22%** |

Figure 4: The feasible cumulative rewards (left two columns) and constraint violation rate (right two columns). We denote the results for different agents with $z\_0$ and $z\_1$. From top to bottom, the environments are Blocked Antwall, Blocked Half-Cheetah, Blocked Walker, and Blocked Swimmer.

Figure 5 displays the performance of MMICRL in both continuous and discrete settings. We observe that it can successfully recover multiple constraints corresponding to different types of agents. The errors made at the beginning of training do not have a significant impact on the overall performance, which demonstrates the robustness of MMICRL.

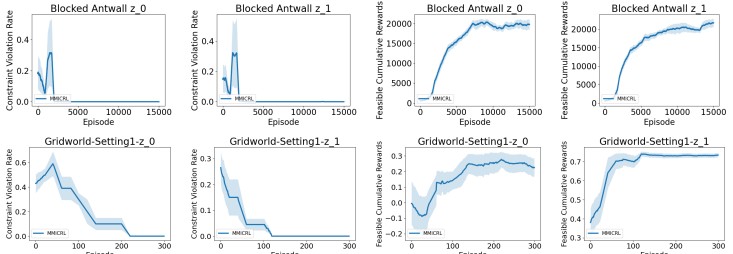

Figure 5: The constraint violation rate (left two columns), and feasible cumulative rewards (right two columns) in Blocked Ant and Gridworld-Setting1( Appendix C.3 shows complete results).

## 5.4 Empirical Evaluations in Realistic Environments

To demonstrate the generalization capability of the model, we conducted some preliminary studies in a realistic autonomous driving environment. This environment is constructed by utilizing the HighD dataset (For more details, check [7] and [9]). For extracting features of cars and roads, we use the features collector from Commonroad RL [39]. The constraint that we are interested in are 1) Car distance $\geq 20$m **(agent 0)** and 2) Car distance $\geq 40$m **(agent 1)**. Figure 6 shows the distribution of car distance in expert demonstrations for agent 0 and agent 1. We aim to investigate whether the MMICRL algorithm can differentiate between two different types of cars based on distance constraints.

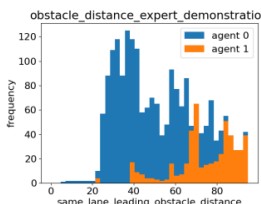

Figure 6: Car distance distribution.

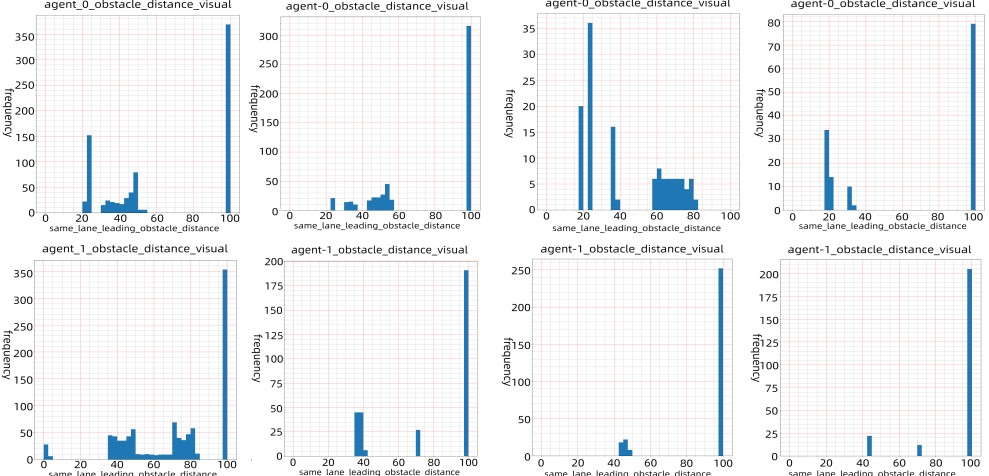

Figure 7: MMICRL performance in a realistic environment. We test four different settings (each column refers to one setting.). We visualize the car distance in the states generated by the imitation policies learned under the inferred constraints for agent 0 (upper row) and 1(lower row).

In the four settings shown in Figure 7, we find that in the first row, the observed distances of the first type of car are all above 20 meters, while in the second row, the distances of the other type of car are mostly above 40 meters. This suggests that the MMICRL algorithm shows promising preliminary results in this context. In the future, we will further optimize the algorithm to achieve a lower constraint violation rate and higher effective rewards in a wider range of real-world environments.

## 6 Limitations

**Omitting Agent Interactions**: MMICRL does not consider the interactions between agents or how they can potentially collaborate or compete to satisfy a joint constraint. Future research could extend the game theory to ICRL for implementing multi-agent constraint inference.

**Experiment in Virtual Environment.** We evaluate our algorithm in virtual games rather than real-world applications (e.g., autonomous driving). This is due to the lack of an ICRL benchmark for a mixture of experts. Future work can explore the application of MMICRL in realistic scenarios.

## 7 Conclusion

In this work, we introduce the MMICRL algorithm to differentiate multiple constraints corresponding to various types of agents. MMICRL incorporates unsupervised constraint inference, agent-specific constraint inference, and multi-modal policy optimization. To demonstrate the advantages of our method over other baselines, we investigate whether MMICRL can accurately perform multiple constraint inference in both discrete and continuous environments.

## Acknowledgments and Disclosure of Funding

The work is supported in part by the National Key R&D Program of China under grant No2022ZD0116004, by Shenzhen Science and Technology Program ZDSYS20211021111415025, and by the Start-up Fund UDF01002911 of the Chinese University of Hong Kong, Shenzhen. We acknowledge the funding from the Canada CIFAR AI Chairs program, and the support of the Natural Sciences and Engineering Research Council of Canada (NSERC).

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
