# Appendix

## A    Implementation Details

### A.1    More Information About The Continuous Environment

We provide a detailed description of the continuous environments with constrained settings:

- *Blocked Half-Cheetah.* The agent controls a robot that has an 18-dimensional state space and can execute actions within a 6-dimensional action space. The reward is determined by the distance the robot walks in each time step, along with a penalty based on the magnitude of the input action. The game ends when a maximum time step is reached. We define two types of constraints that block the region with X-coordinate $\leq -3$ and X-coordinate $\geq 3$, so robots are only allowed to move in the area with X-coordinate between $-\infty$ and -3 under the $1^{st}$ constraint or between 3 and $\infty$ under the $2^{nd}$ constraint.

- *Blocked Ant.* The agent controls a robot that has a 113-dimensional state space and can execute actions within an 8-dimensional action space. The rewards are determined by the distance to the origin and a healthy bonus that encourages the robot to stay balanced. The game ends when a maximum time step is reached. We also define two types of constraints that block the region with X-coordinate $\leq -3$ and X-coordinate $\geq 3$, so robots are only allowed to move in the part with X-coordinate between $-\infty$ and -3 under the $1^{st}$ constraint or between 3 and $\infty$ under the $2^{nd}$ constraint.

- *Blocked Walker.* The agent controls a robot that has an 18-dimensional state space and can execute actions within a 6-dimensional action space. The reward is determined by the distance the robot walks in each time step, along with a penalty based on the magnitude of the input action. The game ends when the robot loses its balance or reaches a maximum time step. We define two types of constraints that block the region with X-coordinate $\leq -0.1$ and X-coordinate $\geq 0.1$, so robots are only allowed to move in the area with X-coordinate between $-\infty$ and -0.1 under the $1^{st}$ constraint or between 0.1 and $\infty$ under the $2^{nd}$ constraint.

- *Blocked Swimmer.* The agent controls a robot that has a 10-dimensional state space and can execute actions within a 2-dimensional action space. The reward is determined by the distance the robot walks in each time step, along with a penalty based on the magnitude of the input action. The game ends when the robot reaches a maximum time step. We define two types of constraints that block the region with X-coordinate $\leq -0.01$ and X-coordinate $\geq 0.01$, so robots are only allowed to move in the area with X-coordinate between $-\infty$ and -0.01 under the $1^{st}$ constraint or between 0.01 and $\infty$ under the $2^{nd}$ constraint.

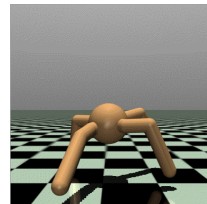 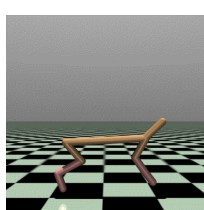 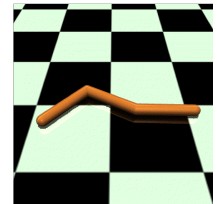 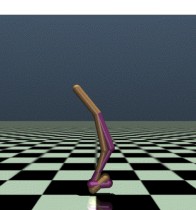

Figure A.1: Four robots: Ant, half-cheetah, swimmer, and walker.

37th Conference on Neural Information Processing Systems (NeurIPS 2023).

## A.2 Hyper-Parameters

**In the MuJoCo environments**, we set 1) the batch size of ME-PPO-Lag [1] to 64, 2) the size of the hidden layer to 64, and 3) the number of hidden layers for the policy function, the value function, and the cost function to 3. We decide on other parameters, including the learning rate of both the policy models and mixture constraint models, by following some previous work [2]. The random seeds of MuJoCo environments are 123, 321, and 666. **In the Gridworld environments**, we set 1) the size of the hidden layer to 64, and 2) the number of hidden layers for the cost function to 3. We determine other parameters, including the learning rate of constraint models, by following [1]. The random seeds of Gridworld environments are 123, 321, and 666.

## A.3 Experiment Equipment

We run the experiments by utilizing multiple kinds of GPUs, including RTX 3060 with 12 GB memory and RTX 3090 with 24 GB memory. We used machines with 12 GB of memory for training the MMICRL models. Given the aforementioned resources, running one seed in the Gridworld environments and the MuJoCo environments takes 1-2 hours and 24-48 hours respectively.

## B Proof of Proposition 4.1

Let's consider an optimization problem in the form of:

$$\text{minimize} - \alpha_1 \mathcal{H}(\pi(\tau)) + \alpha_2 \mathcal{H}(\pi(\tau|z)) \tag{1}$$

$$\text{subject to} \int \pi(\tau|z) f_z(\tau) \mathrm{d}\tau = \frac{1}{N} \sum_{\tau \in \mathcal{D}_z} f_z(\tau) \tag{2}$$

$$\int \pi(\tau|z) \mathrm{d}\tau = 1 \tag{3}$$

$$\int \pi(\tau|z) \log \phi_\omega(\tau|z) \mathrm{d}\tau \geq \epsilon \tag{4}$$

The Lagrange function can be written as:

$$
\begin{aligned}
&L[p(\tau), \lambda_0, \lambda_1, \lambda_2)] \\
=&\alpha_1 \int \pi(\tau) \log \pi(\tau) \mathrm{d}\tau - \alpha_2 \int \pi(\tau|z) \log \pi(\tau|z) \mathrm{d}\tau + \lambda_0 \Big[ \int \pi(\tau|z) f_z(\tau) \mathrm{d}\tau - \tilde{f}_z \Big] + \\
&\lambda_1 \Big[ \int \pi(\tau|z) \mathrm{d}\tau - 1 \Big] - \lambda_2 \Big[ \int \pi(\tau|z) \log \phi_\omega(\tau|z) \mathrm{d}\tau - \epsilon \Big]
\end{aligned}
$$

since $\mathcal{H}(\pi(\tau)) = \sum_z p(z) \mathcal{H}(\pi(\tau|z))$, we can further derive:

$$\mathcal{H}(\pi(r|z)) = \mathbb{E}_{z \sim p(z), (r) \sim \pi(r|z)}(-\log(\pi(r|z)) \tag{5}$$

$$= -\mathbb{E}_{z \sim p(z), (r) \sim \pi(r|z)} \log \left( p(z|r) \frac{\pi(r)}{p(z)} \right) \tag{6}$$

$$= -\mathbb{E}_{z \sim p(z), (r) \sim \pi(r|z)} \log(p(z|r)) - \mathbb{E}_{z \sim p(z), r \sim \pi(r|z)} \log(\pi(r)) + \mathbb{E}_{z \sim p(z)} \log(p(z)) \tag{7}$$

$$= -\mathbb{E}_{z \sim p(z), r \sim \pi(r|z)} \log(p(z|r)) + \mathcal{H}(\pi(r)) - \mathcal{H}(z) \tag{8}$$

$$\mathcal{H}(\pi(\tau)) = \mathcal{H}(\pi(\tau|z)) + \mathbb{E}_{z \sim p(z), \tau \sim \pi(\tau|z)} \Big[ \log(p(z|\tau)) \Big] + \mathcal{H}(z) \tag{9}$$

Substitute it back in the Lagrange function, we get:

$$L[p(\tau), \lambda_0, \lambda_1, \lambda_2)]$$
$$= (\alpha_1 - \alpha_2) \int \pi(\tau|z) \log \pi(\tau|z) \mathrm{d}\tau + \alpha_1 \mathbb{E}_{z \sim p(z), \tau \sim \pi(\tau|z)} \Big[ \log(p(z|\tau)) \Big] + \alpha_1 \mathcal{H}(z) + \lambda_0 \Big[ \int \pi(\tau|z) f_z(\tau) \mathrm{d}\tau - \tilde{f}_z \Big] +$$
$$\lambda_1 \Big[ \int \pi(\tau|z) \mathrm{d}\tau - 1 \Big] - \lambda_2 \Big[ \int \pi(\tau|z) \log \phi_\omega(\tau|z) \mathrm{d}\tau - \epsilon \Big]$$

The Lagrange dual function is:

$$g(\lambda_0, \lambda_1, \lambda_2) = \inf_{\pi(\tau)} L[\pi(\tau|z), \lambda_0, \lambda_1, \lambda_2)]$$
$$= \inf_{\pi(\tau|z)} \int \Big[ \pi(\tau|z) \Big( (\alpha_1 - \alpha_2) \log \pi(\tau|z) + \alpha_1 \mathbb{E}_{z \sim p(z)} [\log(p(z|\tau))] + \lambda_0 f(\tau)_z + \lambda_1 - \lambda_2 \log \phi_\omega(\tau|z) \Big) \Big] \mathrm{d}\tau$$
$$- \lambda_0 \tilde{f}_z - \lambda_1 + \lambda_2 \epsilon + \alpha_1 \mathcal{H}(z)$$

The minimum arrives when:

$$(\alpha_1 - \alpha_2) \log \pi(\tau|z) + \alpha_1 \mathbb{E}_{z \sim p(z)} [\log(p(z|\tau))] + \lambda_0 f(\tau)_z + \lambda_1 - \lambda_2 \log \phi_\omega(\tau|z) = 0 \qquad (10)$$

Equivalently:

$$\pi(\tau|z) = \exp \left[ \frac{-\alpha_1 \mathbb{E}_{z \sim p(z)} [\log(p(z|\tau))] - 1 - \lambda_0 f(\tau) - \lambda_1 + \lambda_2 \log \phi_\omega(\tau|z)}{\alpha_1 - \alpha_2} \right] \qquad (11)$$
$$= \frac{\exp \left[ \frac{-\alpha_1 \mathbb{E}_{z \sim p(z)} [\log(p(z|\tau))] - \lambda_0 f(\tau)}{\alpha_1 - \alpha_2} \right] \phi(\tau|z)^{\frac{\lambda_2}{\alpha_1 - \alpha_2}}}{Z_{\mathcal{M}_c}}$$

## C   More Experimental Results

### C.1   Complementary Results In The Discrete Environment

After analyzing Table C.1 and Figure C.1, it is evident that the B2CL, MEICRL, and InfoGAIL-ICRL methods exhibit unsatisfactory performance. These methods demonstrate a low effective cumulative reward across most environments, accompanied by a high constraint violation rate. Although MMICRL-LD shows a notable improvement, its performance remains mediocre in environments involving three types of agents. In contrast, the MMICRL algorithm consistently achieves good performance across all settings.

### C.2   Complementary Results In The Continuous Environment

Table C.2 presents the mean±std results of all algorithms in Mujoco. We calculated the average for all results and marked those with high values. Figure C.2 depicts the distribution of x-coordinate values among states visited by agents of diverse types across all algorithms in the Blocked Ant, Blocked Half-Cheetah, Blocked Swimmer, and Blocked Walker environments.

### C.3   Complementary Results In The Robust Test

Figure C.3, Figure C.4, Figure C.5 and Figure C.6 display the complete test results of all algorithms in both discrete and continuous environments, providing a comprehensive depiction of the robustness of the MMICRL algorithm. It demonstrates the algorithm's capacity to infer and restore incorrect types of data, highlighting its exceptional adaptability and resilience.

Table C.1: Evaluating performance in Gridworld. We employ "/" to separate the results for various agent types. We present the mean±std results calculated over 20 runs for each random seed.

| Method | Setting 1 | Setting 2 | Setting 3 | Setting 4 |
|---|---|---|---|---|
| **Feasible Cumulative Rewards** | | | | |
| B2CL | $0.24 \pm 0.40/0.55 \pm 0.20$ | $0.61 \pm 0.05 / 0.26 \pm 0.17$ | $0.42 \pm 0.18/0.12 \pm 0.21$ | $0.01 \pm 0.15 /0.28 \pm 0.17/-0.20 \pm 0.02$ |
| MEICRL | $0.46 \pm 0.18/0.43 \pm 0.20 /$ | $0.36 \pm 0.15 / 0.54 \pm 0.06$ | $0.38 \pm 0.18 / 0.45 \pm 0.16$ | $0.05 \pm 0.06/0.53 \pm 0.06/-0.16 \pm 0.03$ |
| InfoGAIL-ICRL | $0.19 \pm 0.17/ - 0.29 \pm 0.11$ | $-0.09 \pm 0.05 / -0.12 \pm 0.04$ | $\mathbf{0.61 \pm 0.24/0.62 \pm 0.23}$ | $-0.32 \pm 0.03 /0.13 \pm 0.10/-0.54 \pm 0.35$ |
| MMICRL-LD | $0.57 \pm 0.17/0.32 \pm 0.18$ | $0.61 \pm 0.08/0.69 \pm 0.10$ | $0.73 \pm 0.06 / 0.31 \pm 0.21$ | $0.01 \pm 0.13 /0.51 \pm 0.08 /0.33 \pm 0.07$ |
| MMICRL | $\mathbf{0.69 \pm 0.14/0.53 \pm 0.11}$ | $\mathbf{0.67 \pm 0.09 /0.68 \pm 0.10}$ | $0.65 \pm 0.13/0.51 \pm 0.12$ | $\mathbf{0.67 \pm 0.10 /0.54 \pm 0.10 /0.39 \pm 0.03}$ |
| **Constraint Violation Rate** | | | | |
| B2CL | $66\% \pm 24\%/33\% \pm 24\%$ | $0\% \pm 0\% /49\% \pm 21\%$ | $33\% \pm 24\%/67\% \pm 24\%$ | $67\% \pm 24\%/33\% \pm 24\%/100\% \pm 0\%$ |
| MEICRL | $30\% \pm 21\% / 33\% \pm 24\%$ | $35\% \pm 19\%/8\% \pm 6\%$ | $33\% \pm 24\%/25\% \pm 17\%$ | $100\% \pm 0\%/0\% \pm 0\%/100\% \pm 0\%$ |
| InfoGAIL-ICRL | $\mathbf{0\% \pm 0\%/0\% \pm 0\%}$ | $91\% \pm 5\%/91\% \pm 6\%$ | $25\% \pm 18\%/33\% \pm 23\%$ | $0\% \pm 0\%/47\% \pm 16\%/0\% \pm 0\%$ |
| MMICRL-LD | $20\% \pm 14\%/40\% \pm 19\%$ | $0\% \pm 0\%/9\% \pm 6\%$ | $0\% \pm 0\%/33\% \pm 24\%$ | $67\% \pm 18\%/0\% \pm 0\%/14\% \pm 10\%$ |
| MMICRL | $\mathbf{0\% \pm 0\%/0\% \pm 0\%}$ | $\mathbf{0\% \pm 0\%/3\% \pm 2\%}$ | $\mathbf{10\% \pm 7\%/4\% \pm 3\%}$ | $\mathbf{0\% \pm 0\%/0\% \pm 0\%/0\% \pm 0\%}$ |
| **Desitination Reaching Rate** | | | | |
| B2CL | $100\% \pm 0\% / 100\% \pm 0\%$ | $67\% \pm 15\% / 100\% \pm 0\%$ | $100\% \pm 0\% / 100\% \pm 0\%$ | $100\% \pm 0\% / 100\% \pm 0\% / 100\% \pm 0\%$ |
| MEICRL | $100\% \pm 0\% / 100\% \pm 0\%$ | $100\% \pm 0\% / 100\% \pm 0\%$ | $100\% \pm 0\% / 100\% \pm 0\%$ | $100\% \pm 0\% / 100\% \pm 0\% / 100\% \pm 0\%$ |
| InfoGAIL-ICRL | $33\% \pm 47\% / 0\% \pm 0\%$ | $33\% \pm 47\% / 0\% \pm 0\%$ | $100\% \pm 0\% / 100\% \pm 0\%$ | $0\% \pm 0\% / 100 \pm 0\% / 33\% \pm 47\%$ |
| MMICRL-LD | $100\% \pm 0\% / 100\% \pm 0\%$ | $100\% \pm 0\% / 100\% \pm 0\%$ | $100\% \pm 0\% / 100\% \pm 0\%$ | $67\% \pm 47\% / 100\% \pm 0\% / 100\% \pm 0\%$ |
| MMICRL | $100\% \pm 0\% / 100\% \pm 0\%$ | $100\% \pm 0\% / 100\% \pm 0\%$ | $100\% \pm 0\% / 100\% \pm 0\%$ | $100\% \pm 0\% / 100\% \pm 0\% / 100\% \pm 0\%$ |

Figure C.1: The feasible cumulative rewards (left two columns of the first three rows and second-to-last row) and constraint violation rate (right two columns of the first three rows and last row). We denote the results for different agents with $z\_0$, $z\_1$, and $z\_2$. From top to bottom, the environments are four Gridworld settings.

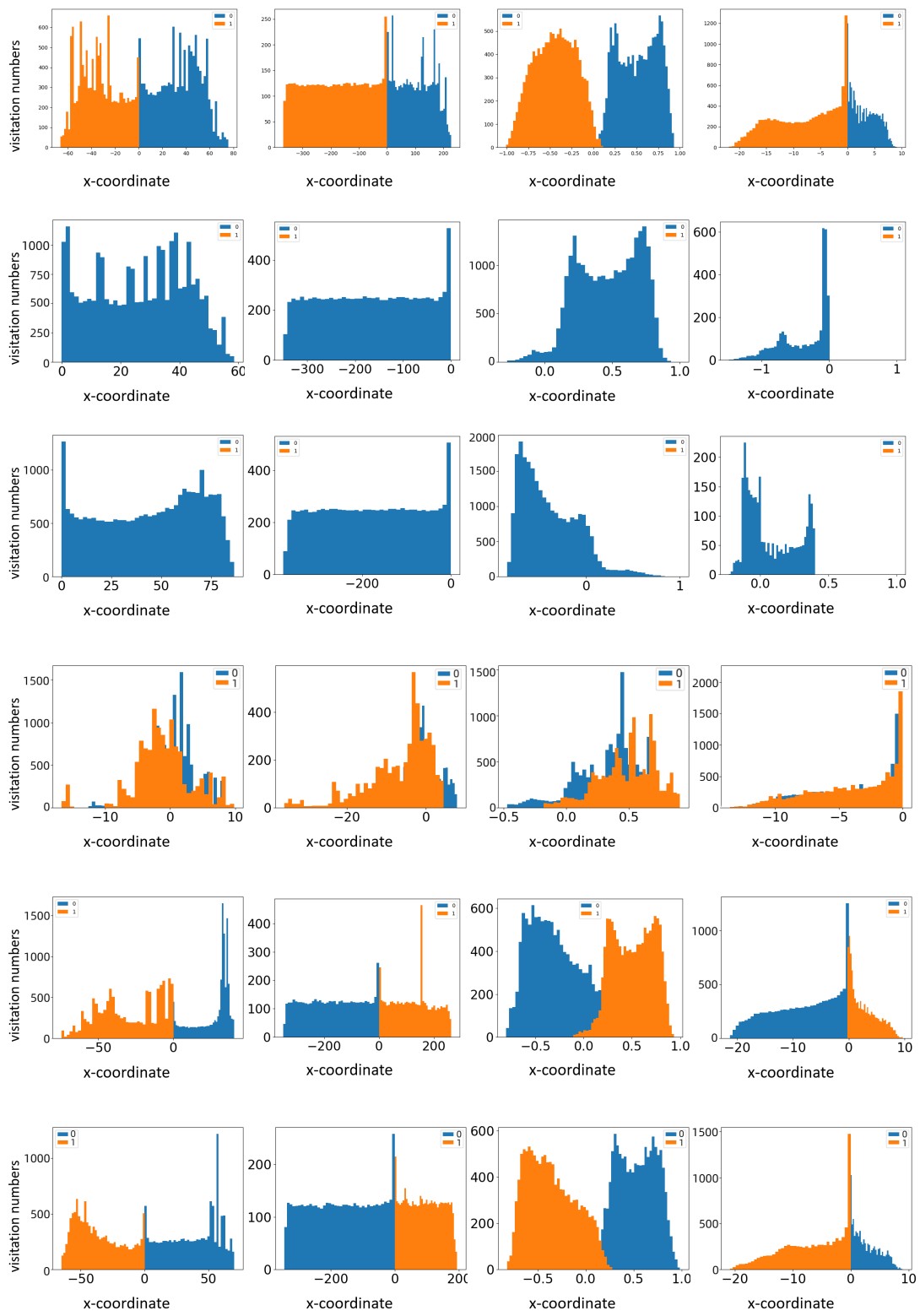

Figure C.2: The above four environments are Blocked Ant, Blocked Half-Cheetah, Blocked Swimmer, and Blocked Walker in order from left to right. The first row showcases the expert demonstration, followed by the results of B2CL, MEICRL, InfoGAIL-ICRL, MMICRL-LD, and MMICRL algorithms. The Blue and orange areas respectively represent the activity trajectories of two types of agents. B2CL and MEICRL can recover only one constraint, so we utilize only one color.

Table C.2: MuJoCo testing performance. We report the average feasible rewards and the constraint violation rate in 100 runs.

| Method | Blocked Half-Cheetah | Blocked Ant | Blocked Swimmer | Blocked Walker |
|---|---|---|---|---|
| **Feasible Cumulative Rewards** | | | | |
| B2CL | 50.29±0.75 / 2120.05±1460.02 | 13322.35±4764.12 / 3112.40±2195.42 | 328.30±217.74 / 0.74±0.20 | 21.90±0.88 / 78.84±0.0 |
| MEICRL | 48.54±0.61 / 3451.08±1698.80 | 17983.07±2681.16 / 3307.97±2340.16 | 201.07±256.70 / 0.83±0.36 | 26.07±3.02 / 73.28±0.0 |
| InfoGAIL-ICRL | 222.44±184.39 / 132.50±65.88 | 372.78±830.08 / 111.33±679.35 | 23.79±19.93 / 0.79±0.24 | 21.45±2.16 / 1579.31±156.02 |
| MMICRL-LD | 4315.32±1513.39 / 2555.59±925.74 | 18158.37±756.88/ 22099.84±824.30 | 265.93±240.60 / 609.83±208.30 | 901.54±544.32 / 511.82±244.21 |
| MMICRL | **6120.68±257.35 / 3064.62±591.24** | **21285.86±1671.92 / 21694.22±889.63** | **407.10±358.78 / 648.11±182.50** | **897.93±476.37 / 1527.49±529.65** |
| **Constraint Violation Rate** | | | | |
| B2CL | 100%±0% / 67%±24% | 33%±24% / 67%±24% | 62%±12% / 100%±0% | 100%±0% / 100%±0% |
| MEICRL | 100%±0% / 50%±25% | 0%±0% / 67%±24% | 79%±14% / 100%±0% | 100%±0% / 100%±0% |
| InfoGAIL-ICRL | 25%±16% / 93%±5% | 34%±18% / 37%±19% | 73%±12% / 100%±0% | 100%±0% / 0%±0% |
| MMICRL-LD | 33%±24% / 34%±24% | **0%±0% / 0%±0%** | 71%±16% / 34%±23% | 52%±25% / 50%±25% |
| MMICRL | **0%±0% / 0%±0%** | **0%±0% / 0%±0%** | **55%±23% / 28%±19%** | **31%±22% / 25%±22%** |

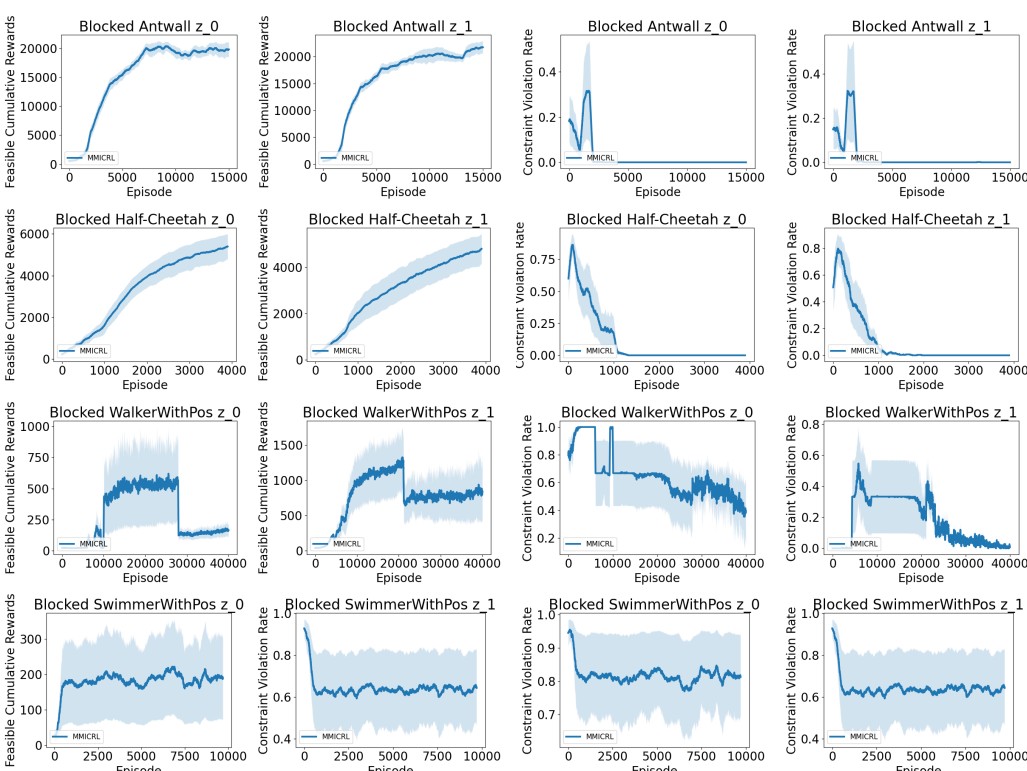

Figure C.3: The feasible cumulative rewards (left two columns) and constraint violation rate (right two columns). We denote the results for different agents with $z\_0$ and $z\_1$. From top to bottom, the environments are Blocked Antwall, Blocked Half-Cheetah, Blocked Walker, and Blocked Swimmer.

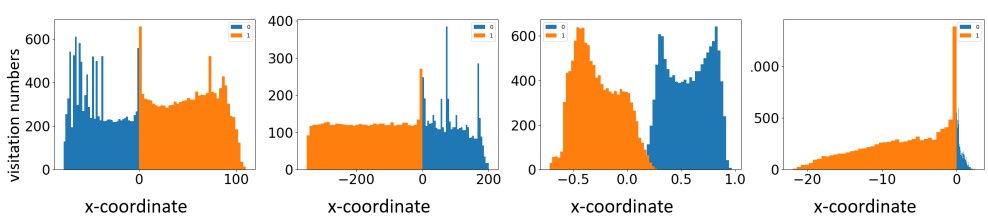

Figure C.4: From left to right, the four environments are Blocked Ant, Blocked Half-Cheetah, Blocked Swimmer, and Blocked Walker.

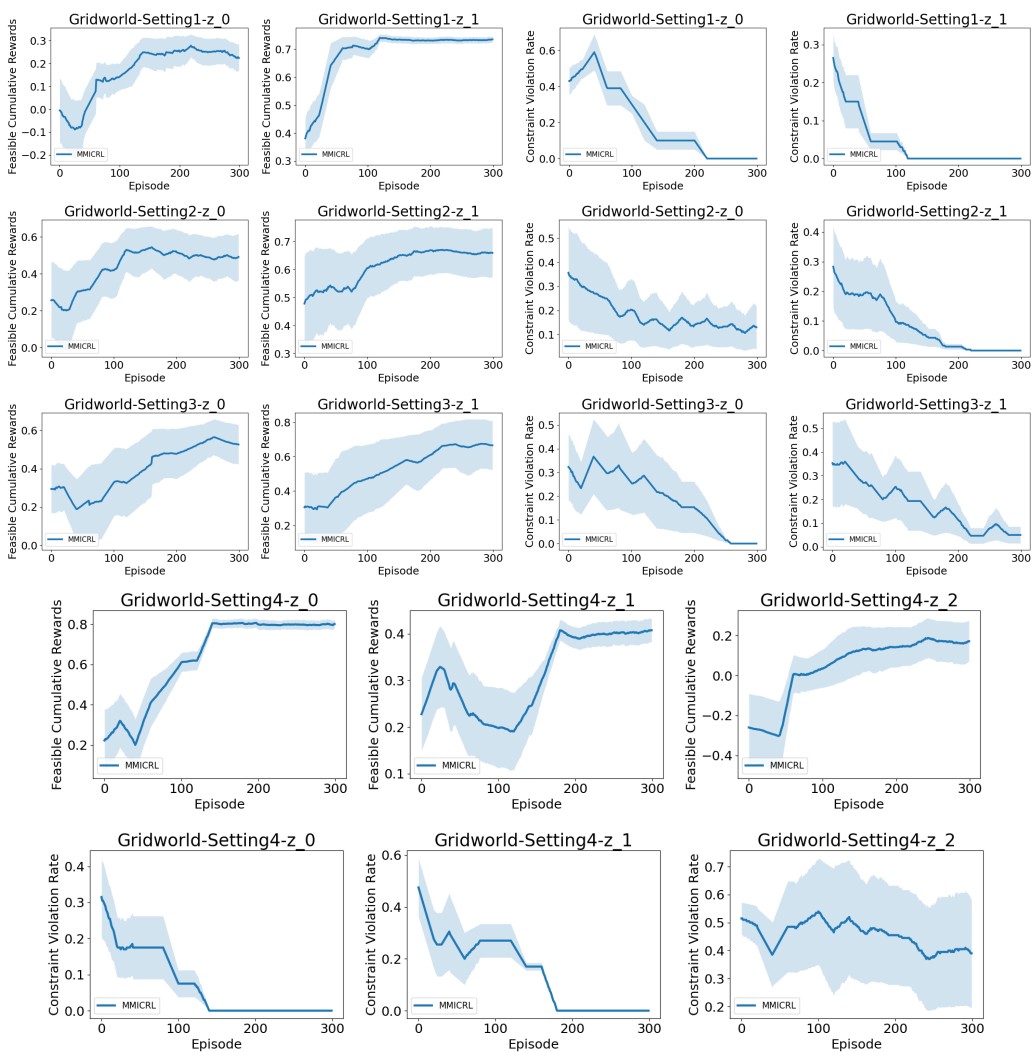

Figure C.5: The feasible cumulative rewards (left two columns of the first three rows and second-to-last row) and constraint violation rate (right two columns of the first three rows and last row). We denote the results for different agents with $z\_0$, $z\_1$, and $z\_2$.

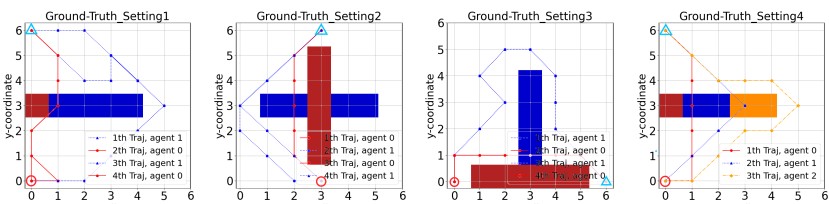

Figure C.6: The recovered trajectories for the 4 settings (from top to bottom) in the robust test.

# D    Supplementary Experiment

## D.1    Impact of probing vectors

In this experiment, our objective was to investigate the impact of the number of probing vectors in Blocked Half-Cheetah. We maintained all other parameters constant and solely manipulated the number of probing vectors. Figure D.2 demonstrates that there is no discernible effect when the number of probing vectors is below 10. However, a notable effect is observed when the number is increased to 30. The experimental results indicate that we can enhance the algorithm's effectiveness within a specific range by identifying more feature points to differentiate different types of agents. These findings highlight the significance of finding an optimal balance in the number of probing vectors for improved performance.

## D.2    Contrastive Experiments for Optimizing Objectives

This experiment serves to highlight the importance of $\mathcal{H}[\pi(\tau|z)]$ in the optimization objective, as outlined in Equation 8 of the main text. By integrating $\mathcal{H}[\pi(\tau|z)]$, we can effectively regulate the variance of the policy distribution, specifically for a particular type of agent. Consequently, we can acquire trajectories that pertain to diverse agent types. Figure D.1 vividly illustrates the consequences of removing $\mathcal{H}[\pi(\tau|z)]$ within the Blocked Half-Cheetah environment and Grid-

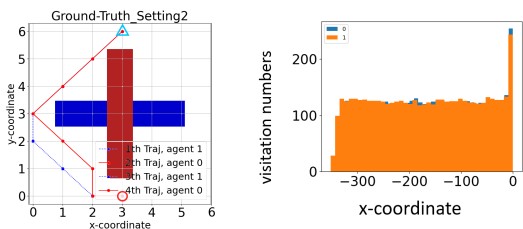

Figure D.1: The effect of removing $\mathcal{H}[\pi(\tau|z)]$ factor in both discrete and continuous environments.

world setting2. Without it, we are confined to generating trajectories exclusively for one agent type, thereby restricting our ability to discern and accommodate constrained types. To enhance the performance of the MMICRL algorithm, striking a delicate equilibrium between this factor and the maximum entropy policy optimization becomes important.

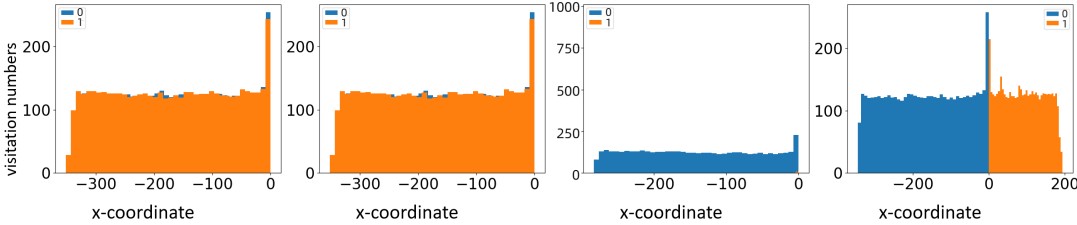

Figure D.2: The number of probing vectors is 1, 5, 10, and 30 from left to right.

## D.3    Robustness Tests for Agent Types

In this small experiment, we want to observe the importance of Setting an Upper Bound on the Number of Agent Types. In Figure D.3, the number of agent types (hyper-parameters) in MMICRL is larger than the number of actual agent types. Despite this discrepancy, our algorithm continues to exhibit satisfactory performance. In Figure D.4, the number of agent types (hyper-parameters) in MMICRL is smaller than the actual number of agent types. The inference constraint error occurs for one specific agent type, and repeated experiments have failed to recover the correct outcome. Table D.1 provides the specific result.

## D.4    Experiments with a Larger Variety of Agent Types

To test the generalization of MMICRL, we set the number of agent types to 4. Figure D.5 and Table D.1 present the good performance of MMICRL.

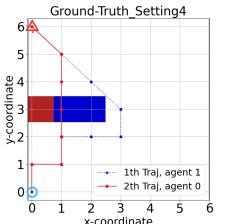 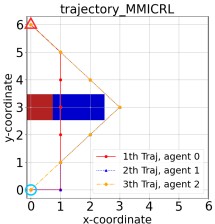 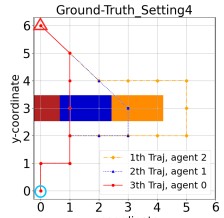 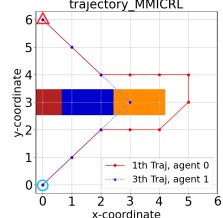

Figure D.3: The ground truth (left) has two types of agents while MMICRL (right) set the agent types to three (i.e., $|\mathcal{Z}| = 3$ is an upper bound).

Figure D.4: The ground truth (left) has three types of agents while MMICRL (right) set the agent types to three (i.e., $|\mathcal{Z}| = 3$ is smaller than the real number).

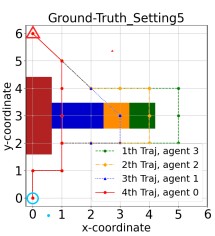 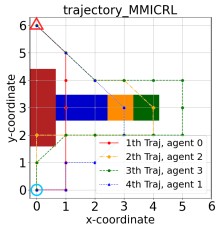

Figure D.5: The ground truth (left) has two types of agents while MMICRL (right) set the agent types to three (i.e., $|\mathcal{Z}| = 3$ is an upper bound).

Table D.1: Supplementary Result

| Experiment setting | Feasible Cumulative Rewards | Constraint Violation Rate |
|---|---|---|
| 2 GT and 3 preset | 0.54 | 0% |
| 3 GT and 2 preset | 0.25 | 33% |
| 4 GT and 4 preset | 0.7 | 0% |