# OpenReview forum: "Multi-Modal Inverse Constrained Reinforcement Learning from a Mixture of Demonstrations"
_NeurIPS.cc/2023/Conference — NeurIPS 2023 poster_

### Official Review · Reviewer_Gwnr · 2023-06-22

**Soundness:** 3 good
**Presentation:** 4 excellent
**Contribution:** 3 good
**Rating:** 8
**Confidence:** 4

**Summary:**

The paper proposes the algorithm Multi-Modal Inverse Constrained Reinforcement Learning (MMICRL) for imitation learning mixture of expert demonstrations with various constraints. The algorithm includes agent identification, agent-specific constraint inference and multi-modal policy optimization. The problem is challenging, the proposed method seems to be novel, and the simulation experiments show the effectiveness of the proposed method.


**Strengths:**

I think the problem of multi-modal learning from a mixture of experts with constraints is indeed a challenging one, and the proposed algorithm has necessary components for handling several difficulties.

The paper has good writing, and the method is described clearly with necessary derivation details. Although lots of mathematical notations are used, it’s good to see intuitive explanations for most terms.

The experiments are evaluated on both discrete and continuous environments. The improvement of the proposed algorithm over baseline methods is significant, although not consistent across tasks.


**Weaknesses:**

Overall the paper has a clear description of the reasons for different design choices, but there are still several places confusing for me, detailed in the Questions section.

For proposition 4.2, please provide the complete conditions for the statement. For the case with a limited number of samples and estimated occupancy measure, what is the change for this statement?

In Table 2, it could be better to provide the constraint violation rates in the expert demonstration data as a reference. Does the demonstration data have zero violation rate? It may be mentioned somewhere in the paper that I missed.

For Fig. 4, is it possible to have zero or lower constraint violation while remaining high rewards for tasks Blocked WalkerWithPos and Blocked SwimmerWithPos.

It could be better to see the method tested on some more realistic cases like vehicle driving dataset, etc.


**Questions:**

From Eq. (9) to (10), why using the contrastive estimation method helps with handling the trade-off of reward maximization and density maximization? The contrastive estimation loss also induces an additional term from the reward.

In Eq. (10), how is the reward of the trajectory estimated conditioned on the current policy? Is there any value function estimation in the policy optimization process?



**Limitations:**

The limitations of the work are well-discussed.

---

> ### Author Rebuttal · Authors · 2023-08-09
>
> Dear Reviewer, we sincerely value your time and effort in evaluating our work. Your insights have been valuable to our work. We have prepared comprehensive responses and clarifications to address each point you raised. We hope these responses can resolve your concerns.
>
> 1. *"For proposition 4.2, please provide the complete conditions for the statement. For the case with a limited number of samples and estimated occupancy measure, what is the change for this statement?"*
>
> **Our response.**  Thanks for your suggestions. We agree that a thorough explanation of proposition 4.2 is necessary. This proposition is in fact citing the Theorem 2 of [15]. The occupancy measures $\rho$ should satisfy the Bellman flow constraints. That's to say:
> \begin{align}
> \sum_a{\rho(s,a)}=P_0(s)+\gamma\sum_{s^{\prime},a}\rho(s^{\prime},a)P_{\mathcal{T}}(s^{\prime}|s,a) \text{ and }{\rho(s,a)}\geq0
> \end{align}
> This constraint is often applied in the dual problem for solving MDP. Since our expert trajectories are assumed to be optimal in solving the primal control problem in MDP, their occupancy measures $\rho^\pi(s,a)$ should generally satisfy the constraints.
>
> [15] Umar Syed, Michael H. Bowling, and Robert E. Schapire. Apprenticeship learning using linear programming. In International Conference on Machine Learning (ICML), pages 1032–1039, 2008.
>
> 2. *" In Table 2, it could be better to provide the constraint violation rates in the expert demonstration data as a reference. Does the demonstration data have zero violation rate? It may be mentioned somewhere in the paper that I missed."*
>
> **Our response.** That's a very good point. We agree that more detailed clarification is needed. In fact, ICRL algorithms typically focus on 1) soft constraints, which demand constraint satisfaction in expectation (i.e., violation rate > 0), and 2) hard constraints requiring absolute constraint satisfaction (i.e., violation rate = 0). While soft constraints are particularly useful in stochastic environments, they're challenging to estimate. For simplicity, we've chosen to focus on hard constraints, which exhibit a zero violation rate in the expert demonstration, in this study.
>
> 3. *"For Fig. 4, is it possible to have zero or lower constraint violation while remaining high rewards for tasks Blocked WalkerWithPos and Blocked SwimmerWithPos."*
>
> **Our response.**  In our experiments, we managed to attain a low constraint violation in **certain** random seeds, but not in **all** of them. The reported results represent **averages** across all tested seeds. Consistently achieving a low constraint violation rate is challenging in both environments. Similar findings are illustrated in Figure 2 of [7], where the MECL (ICRL) algorithm fails to uphold a zero constraint violation rate in the Walker and Swimmer environments. Notably, their experiments focus on a single type of agent, and our task is even more challenging. Future research will be required to make further progress in this regard.
>
> [7] Guiliang Liu, Yudong Luo, Ashish Gaurav, Kasra Rezaee, and Pascal Poupart. Benchmarking constraint inference in inverse reinforcement learning. In International Conference on Learning Representations (ICLR), 2023.
>
> 4.*" It could be better to see the method tested on some more realistic cases like vehicle driving dataset, etc."*
>
> **Our response.** In Section 5.4 Limitations, we acknowledged the absence of a realistic benchmark to test our MMICRL algorithm. We notice a recent study ([Ref.7]) provides an environment tailored for autonomous driving tasks, albeit **limited to a single agent type**. A possible route is extending this environment to include multiple agent types and deriving corresponding constraints. Our methodology and findings are elaborated in the attached PDF.
>
> 5.*"From Eq. (9) to (10), why using the contrastive estimation method helps with handling the trade-off of reward maximization and density maximization? The contrastive estimation loss also induces an additional term from the reward."*
>
> **Our response.** This is an excellent question. Empirically, we conducted an **ablation study** in our experiment (see MMICRL-LD) by omitting our contrastive estimation approach and using objective (9) directly for policy updates. We noticed sub-optimal control performance in our early experiments, which **motivated** us to explore alternative methods. The contrastive estimation technique is a straightforward, effective method for learning feature representations. Compared to directly applying log density, the information-theoretic InfoNCE loss, which transforms **multi-label-classification** into **positive-negative samples differentiation**, serves as a more applicable loss for learning differentiable policies. This method has already shown stable performance in more complex applications.
>
> 6.*"In Eq. (10), how is the reward of the trajectory estimated conditioned on the current policy? Is there any value function estimation in the policy optimization process?"*
>
> **Our response.** ICRL typically assumes a known reward function[4]. In our MMICRL, we assume all experts share the same reward, and explain their distinct behaviors by different constraints. To estimate the cumulative rewards in a trajectory, we follow [4] and utilize of PPO-Lag (Proximal Policy Optimization Lagrange) for policy optimization.
>
> In PPO-Lag, we have an estimation of the value function. The algorithm learns both a reward-based value function and a cost-based value function for calculating expected rewards and costs for a policy.
>
> [4] Shehryar Malik, Usman Anwar, Alireza Aghasi, and Ali Ahmed. Inverse constrained reinforcement learning. In International Conference on Machine Learning (ICML), pages 7390–7399, 2021.

---

> > ### Comment · Reviewer_Gwnr · 2023-08-16
> > **Response to rebuttal**
> >
> > I would like to thank the authors for providing detailed explanation. It answers my questions and I have no further problem at present. I would suggest the authors to add these clarification details to the modified paper.

---

> > > ### Author Response · Authors · 2023-08-17
> > >
> > > Dear Reviewer. Thanks for your comments. We appreciate your time and effort, and we will make sure all these clarification details are included in the modified paper.

---

### Official Review · Reviewer_ieEm · 2023-06-27

**Soundness:** 3 good
**Presentation:** 3 good
**Contribution:** 3 good
**Rating:** 6
**Confidence:** 2

**Summary:**

The paper introduces a new algorithm called Multi-Modal Inverse Constrained Reinforcement Learning (MMICRL) to address the challenge of recovering multiple underlying constraints from a mixture of trajectories demonstrated by different types of expert agents. The algorithm utilizes a flow-based density estimator to identify experts from the demonstration data and infer agent-specific constraints. MMICRL then employs a novel multi-modal constrained policy optimization objective to imitate expert policies, considering both conditioned and unconditioned policy entropy. The approach is further enhanced using contrastive learning to improve robustness. Experimental results in discrete and continuous environments demonstrate that MMICRL outperforms other baseline algorithms in terms of constraint recovery and control performance.

**Strengths:**

The paper introduces a new algorithm, MMICRL, which addresses the challenge of recovering multiple constraints from a mixture of trajectories demonstrated by different types of expert agents. This is a novel and important problem in the field of Inverse Reinforcement Learning (IRL). MMICRL utilizes a flow-based density estimator to identify expert agents from demonstration data. This approach allows for unsupervised expert identification and contributes to the ability to infer agent-specific constraints accurately. The incorporation of the policy optimization objective into the contrastive learning framework enhances the robustness of the algorithm. This addition strengthens the learning process and improves the overall performance of MMICRL.

The paper presents extensive experiments conducted in both discrete and continuous environments to evaluate the performance of MMICRL. The experimental results demonstrate that MMICRL outperforms other baseline algorithms in terms of constraint recovery and control performance, indicating the effectiveness and superiority of the proposed approach.

**Weaknesses:**

I am not an expert in IRL, but I know some RL algorithms [1] make use of mixture models and their modelling methods can even be applied in other fields, like computer vision[2]. I am wondering if is it possible to make these modelling methods in IRL and what is the advantage of MMICRL compared with these methods.

[1] Ren, Jie, et al. "Probabilistic mixture-of-experts for efficient deep reinforcement learning." arXiv preprint arXiv:2104.09122 (2021).

[2] Xia, Xiaobo, et al. "Pluralistic image completion with gaussian mixture models." Advances in Neural Information Processing Systems 35 (2022).

**Questions:**

N/A

---

> ### Author Rebuttal · Authors · 2023-08-09
>
> Dear Reviewer, we sincerely value the time and effort you have devoted to evaluating our work. To address each point you raised, we have prepared comprehensive responses and clarifications. We hope these responses can resolve your concerns.
>
> 1. *"I am not an expert in IRL, but I know some RL algorithms [1] make use of mixture models and their modeling methods can even be applied in other fields, like computer vision[2]. I am wondering if is it possible to make these modeling methods in IRL."*
>
> **Our response.**  We thank the reviewer for providing us with the methods for reference and comparison. Although these works could serve as important relevant works (**We will make sure our related works section can reflect this relevance**), they are not directly applicable to solving ICRL problems. In the following, we elaborate on their difference to our MMICRL and potential extension to ICRL.
>
> a) In the study, "Probabilistic Mixture-of-Experts for Efficient Deep Reinforcement Learning", the authors examined the **unconstrained** forward control problem that learns ** control policy** from **observed** environmental dynamics, such as state and rewards. Their approach employs a Gaussian Mixture Model (GMM) to model a **multi-modal** policy, which they term a mixture of experts.
>
> Several key differences distinguish their approach from ours:
>
> - Our method, MMICRL, addresses an **inverse** optimal control problem, learning **missing** dynamics signals, notably constraint signals, from provided **expert demonstration**.
> - MMICRL operates within a **constrained** Markov Decision Process (MDP), in contrast to their unconstrained model.
> - Instead of modeling multi-modal expert behavior with a single unified policy, MMICRL **differentiates** expert trajectories through agent identification and learns imitation policies **based on the expert type**.
>
> Substituting our PPO-Lag policy representation with a GMM does not account for the contradictory behavior seen in expert demonstrations, particularly in safety-critical situations.
>
> b) In the research, "Pluralistic Image Completion with Gaussian Mixture Models", the primary focus is on constrained pluralistic image completion. The relationship between the sub-procedure and pluralistic results is modeled using a Gaussian Mixture Model (GMM).
>
> Their methodology differs from ours in two main ways:
>
> - MMICRL models the control problem as sequential decisions within a Markov Decision Process (MDP). It's unclear if the task of pluralistic image completion can be mapped similarly to a control problem within an MDP.
>
> - MMICRL learns constraints from expert demonstrations. In contrast, the constraints in pluralistic image completion are either manually provided or derived from the dataset.
>
> 2. *"what is the advantage of MMICRL compared with these methods."*
>
> **Our response.** Since the methods primarily target different problems and settings than MMICRL, direct comparisons may be challenging. However, we argue the necessity of MMICRL based on the following points:
>
> - MMICRL can be concurrently applied to both discrete and continuous environments and various inverse control tasks.
> - MMICRL can estimate agent-specific constraints from a mixture of expert demonstrations, providing an explanation for their contradictory behavior in game contexts.
> - In comparison to other baselines, MMICRL stands out in maintaining a low constraint violation rate while achieving high rewards, demonstrating its effectiveness.
> - Using a GMM policy representation is not a direct substitute for our method. Our approach distinctly excels in differentiating between various types of agents.

---

> > ### Author Response · Authors · 2023-08-21
> >
> > Dear reviewer,
> >
> > We appreciate the time and effort you've devoted to reviewing our paper. We would like to check whether our responses have adequately addressed your concerns. Your feedback is invaluable to us, and we will carefully revise the manuscript and integrate your suggestions to improve its overall quality. Thank you once again for your significant contribution and the time you've invested.
> >
> > Best Regards,
> > Authors of Paper 6041

---

### Official Review · Reviewer_ZL9H · 2023-07-03

**Soundness:** 2 fair
**Presentation:** 2 fair
**Contribution:** 2 fair
**Rating:** 4
**Confidence:** 2

**Summary:**

This paper considers the problem of estimating constraints from a dataset consisting of a mixture of expert trajectories, and proposes an algorithm (MMICRL) for solving that problem.
MMICRL proceeds iteratively, first estimating which trajectories belong to which agent class using a density estimation approach (over state-action occupancy), and second infers constraints on a per-agent class basis (using a MLE constraint inference procedure).
The policy is learned concurrently using an agent-unconditional max entropy, agent-conditional min entropy objective.
Experiments in gridworld and mujoco environments show that MMICRL outperforms baselines as measured by permissible expected return and constraint violation rate.


**Strengths:**

### Originality
- The paper proses what seems to be a novel setting of inverse constrained RL with a mixture of experts
- The combination of methods (unsupervised agent clustering, constraint inference, policy learning) used in MMICRL appears novel, the application of certain methods (state-action occupancy density estimation, contrastive objective) to ICRL seems novel, and some aspects seem entirely novel (in-class min entropy, out-of-class max entropy).
### Quality
Evaluation
- The evaluation of the proposed algorithm is quite strong. Appropriate baselines are considered (given the novel setting they double as ablations), and experiments are well-designed to illustrate the advantages of the approach - the grid world demonstrates the behavior in an interpretable manner, and the mujoco experiments demonstrate the usefulness of function approximation in the setting.
- Experiments are run multiple times over different random seeds, with aggregate performance reported.
- Experiments investigating susceptibility of the algorithm to initial poor performance do a good job of conveying how robust the algorithm is in practice
### Clarity
- The algorithm and experiments are presented fairly clearly, with generally useful / clear figures, a specification of the algorithm, and results clearly summarized / visualized.


**Weaknesses:**

### Significance
- Does constraint inference in the context of a mixture of expert demonstrations makes sense as a problem formulation? If different experts disagree on what is or is not a constraint, then does it make sense to formulate the problem as constraint inference (or would rewards be more appropriate)? The motivation for ICRL is typically that the behavior of experts is often best explained by a simple reward function + a set of constraints [1], but that does not seem to be the case here. The actionable critiques are (1) the paper does not apply the method in a realistic setting and (2) the examples given in the paper of real applications (driving datasets with unlabeled car types) are unrealistic (and relies on partial observability (of the object type) more than a mixture of experts).
### Originality
- Constraint inference in the context of a mixture of expert behavior does seem novel, but the analogous extension for reward learning is not (e.g., [2,3,4])
- MMICRL largely combines existing methods (density estimation, constraint inference, policy optimization) adapting for the multi-modal case (with some novel contributions noted in the strengths)
### Quality
Algorithm
- What can be shown theoretically about the algorithm? For example, does it converge under some assumptions? Discuss of such topics seems missing from the paper
Evaluation
- Related to the point about significance: the evaluation settings are unrealistic. An example with real world data would dramatically improve the quality of the evaluation.
Clarity
- It’s not clear until later in section 4 that the algorithm iteratively performs agent identification / clustering, which makes section 4.1 confusing when first reading through
- Figure 4 could be improved wrt clarity. Perhaps aggregating over environments would help.

[1] Maximum Likelihood Constraint Inference for Inverse Reinforcement Learning \
[2] LiMIIRL: Lightweight Multiple-Intent Inverse Reinforcement Learning \
[3] Dealing with multiple experts and non‑stationarity in inverse reinforcement learning: an application to real‑life problems \
[4] Nonparametric Bayesian Inverse Reinforcement Learning for Multiple Reward Functions


**Questions:**

1. How is evaluation performed given that the agent-class is identified in an unsupervised manner? It seems like the relevant z value must be inferred either heuristically or manually in order to evaluate e.g., constraint violations. If so, this seems like a limitation of the approach, the reason being that identifying which z value to use in a practical setting would be challenging (given the use of CIRL in the first place implies complex / high-dimensional constraints).
2. Is it correct that the method is limited to a setting with a fixed number of agent classes? This appears to be the case based on the algorithm. If so, (1) how robust is the algorithm to the number of classes chosen being incorrect? And (2) this also seems like a major limitation in that often real world data doesn’t cluster into a discrete set (but rather exists along a continuous spectrum).
3. Section 3 has this line: “CMA-MDP assumes the agents are differentiable by examining their policies, implying that different agent types cannot share identical policies”. Would you elaborate on both the differentiability claim and the implication?


**Limitations:**

See weaknesses and questions sections

---

> ### Author Rebuttal · Authors · 2023-08-09
>
> Dear Reviewer, we sincerely value the time and effort you have devoted to evaluating our work. To address each point you raised, we have prepared comprehensive responses. We hope these responses can resolve your concerns.
>
> 1. *"How is evaluation performed given that the agent class is identified in an unsupervised manner? ... identifying which z value to use in a practical setting would be challenging"*
>
> **Our Response.** During the evaluation process, **we must assign the identified agents to a specific type of expert agent.** This is achieved by a majority voting system. To illustrate, let's consider a classified expert dataset, denoted as $D_{z}$ (line 182 of our paper), where 90\% of the trajectories are generated by agent type 1 (this information is available during the evaluation.), so we would label this class as type 1. Subsequently, the corresponding ground-truth constraint related to type 1 would be used to compute the constraint violation rate.
>
> In a **practical application**, such as autonomous driving, we can run our MMICRL algorithm to derive constraints for different agents. To apply these constraints, we must associate identified agents with actual vehicles, which is made by matching the distribution of the identified expert dataset $\mathcal{D}_{z}$ to the distribution observed in the target vehicle. It's crucial to note that this matching wouldn't be feasible without the expert dataset identified by our MMICRL.
>
> 2. *"Is it correct that the method is limited to a setting with a fixed number of agent classes?"*
>
> **Our Response.** In our MMICRL algorithm, the count of agent types of interest, denoted as $|\mathcal{Z}|$, is a hyperparameter. However, it doesn't necessarily match the precise number of expert types but rather serves as an upper limit. Therefore, if the actual expert types are fewer than $|\mathcal{Z}|$, MMICRL will 1) either create additional subclasses for an actual expert type or 2), assigns no expert trajectory to one class, rendering $\mathcal{D}_{z^{''}}=\{\emptyset\}$, which can then be discarded in the subsequent iteration. This behavior doesn't compromise the algorithm's effectiveness. However, if the actual expert types exceed $|\mathcal{Z}|$, implying $|\mathcal{Z}|$ no longer serves as an upper limit, it could lead to sub-optimal performance.
>
> 3. *"How robust is the algorithm to the number of classes chosen being incorrect?"*
>
> **Our Response.** The robustness in MMICRL can be justified by experiment. Please check our additional results in the attached PDF file.
>
> 4. *"This also seems like a major limitation in that often real-world data doesn’t cluster into a discrete set ..."*
>
> **Our Response.** In real life,  we believe there are many examples where agents can be classified into discrete sets. For instance, vehicles of different types, and sports players in different positions. For cases where clear distinctions are difficult to make, we can **discretize the continuous sets** and then use MMICRL to infer the constraints.
>
> 5.*"Section 3 has this line: "CMA-MDP assumes the agents are differentiable by examining their policies ...". Would you elaborate on both the differentiability claim and the implication?"*
>
> **Our Response.** The assumption is consistent with the design of our MMICRL algorithm. If two agent policies are identical, our unsupervised agent identifier would not differentiate them. Consequently, there's no requirement or benefit to regard them as distinct agent types or account for their behaviors using separate constraints.
>
> 6. *"What can be shown theoretically about the algorithm? For example, does it converge under some assumptions? ...".*
>
> **Our response.**  While we acknowledge the **significance of such understanding for ICRL**, theoretical results for ICRL are, to our knowledge, currently **lacking**. A potential route is extending theoretical findings from IRL to ICRL, but enabling this extension remains a complex issue. because:
>
> - The advancements in IRL theory depend on defining feasible sets of rewards. However, defining a feasible set of **constraints** is challenging due to the variability of constraints (e.g., hard, soft, and probabilistic constraints) and the constrained optimization method (e.g., Lagrange method, Interior points). These issues are notably **absent in IRL**, where the forward problem is unconstrained.
>
> - IRL theories usually deal with a simplified setting involving bounded rewards and a tabular environment. In contrast, our work explores environments with a continuous state space and multiple expert policies and demonstrations. The gap between rigorous theoretical understanding and our ICRL setting is substantial.
>
> 7. "If different experts disagree on what is or is not a constraint, then does it make sense to formulate the problem as constraint inference (or would rewards be more appropriate)?"
>
> **Our response.** It is challenging to explain the conflicting behaviors among expert trajectories through reward inference with IRL. For instance, in a situation where 50\% of expert agents move left and the other half move right, IRL would assign near-zero rewards to both actions, considering that each action is deemed sub-optimal 50\% of the time.
>
> Our approach indeed aligns with the motivation for ICRL, as we assume all expert agents adhere to the same known reward function. We account for their disagreements by learning a series of distinct constraints.
>
> 8. *"Concerns about the unrealistic evaluation."*
>
> **Our response.** Section 5.4 Limitations have acknowledged the absence of a realistic benchmark.  A recent study ([Ref.7]) describes an environment tailored for autonomous driving tasks, albeit **limited to a single agent type**. A possible route is extending this environment to include multiple agent types. Our methodologies are elaborated on in the attached PDF.

---

> > ### Author Response · Authors · 2023-08-21
> >
> > Dear reviewer,
> >
> > We appreciate the time and effort you've devoted to reviewing our paper. We would like to check whether our responses have adequately addressed your concerns. Your feedback is invaluable to us, and we will carefully revise the manuscript and integrate your suggestions to improve its overall quality. Thank you once again for your significant contribution and the time you've invested.
> >
> > Best Regards,
> > Authors of Paper 6041

---

### Official Review · Reviewer_KNkP · 2023-07-07

**Soundness:** 4 excellent
**Presentation:** 3 good
**Contribution:** 3 good
**Rating:** 7
**Confidence:** 3

**Summary:**

# Problem Statement
The paper addresses a significant problem in Inverse Constraint Reinforcement Learning (ICRL), which is the assumption that all expert demonstrations follow the same constraints. This assumption is problematic because in real-world scenarios, demonstration data may come from various agents who follow different or even conflicting constraints. Therefore, using a single constraint model to explain the behaviors of diverse agents can lead to inaccuracies.

# Main Contribution
To tackle this issue, the paper introduces the Multi-Modal Inverse Constrained Reinforcement Learning (MMICRL) algorithm. This approach allows imitation policies to capture the diversity of behaviors among expert agents. The paper demonstrates that MMICRL outperforms other baselines in terms of constraint recovery and control performance in both discrete and continuous environments.

# Methedology
The proposed MMICRL estimates multiple constraints corresponding to different types of experts, allowing for a more accurate representation of diverse agent behaviors. It uses a flow-based density estimator for unsupervised expert identification from demonstrations, infers agent-specific constraints, and optimizes a multi-modal policy that minimizes the agent-conditioned policy entropy and maximizes the unconditioned one.

The key steps of MMICRL include unsupervised agent identification, conditional inverse constraint inference, and multi-modal policy update.

1. Unsupervised Agent Identification: MMICRL identifies expert trajectories in an unsupervised manner. It performs trajectory-level identification by estimating an agent-specific density. The algorithm uses a Conditional Flow-based Density Estimator (CFDE) to compute a density using a state-action density estimator. The agent identifier is represented by the softmax representation.

2. Agent-Specific Constraint Inference: Based on the identified expert dataset, the likelihood function can be simplified. The algorithm parameterizes the instantaneous permissibility function with $ω$ and updates the parameters by computing the gradient of the above likelihood function.

3. Multi-Modal Policy Optimization: The policy is trained to maximize cumulative rewards subject to constraint. The objective expands the reward signals with a log-probability term, which encourages the policy to generate trajectories from high-density regions for a specific agent type. The algorithm integrates contrastive learning into policy optimization, helping it to better understand the relationships between agents, their behaviors, and the corresponding expert trajectories.

The algorithm alternates between these steps until the imitation policies reproduce expert trajectories, signifying that the inferred constraints align with the ground-truth constraints.

# Experiments

The paper conducts experiments on the MMICRL algorithm in both discrete and continuous environments, using Constraint Violation Rate and Feasible Cumulative Rewards as evaluation metrics. In discrete environments based on Gridworld, MMICRL successfully identifies various agent types and generates accurate trajectories. In continuous environments based on MuJoCo, MMICRL consistently exhibits lower constraint violation rates and higher cumulative rewards, outperforming other baseline methods. A robustness test shows that MMICRL can recover from initial incorrect agent identifications, demonstrating its ability to accurately model diverse agent preferences and its robustness in handling errors.

**Strengths:**

# Originality and significance
This work poses a unique problem setup of multi-modal ICRL, which nevertheless has substantial real-world impact for applications such as autonomous driving or human-robot interaction. The proposed method is sophisticated and innovative, requiring no supervision other than expert demonstrations.

# Quality
A comprehensive analysis, along with an ablation study, is conducted in comparison to various baselines, effectively validating the efficacy of the proposed method.

**Weaknesses:**

- The number of agent types needs to be preset, and such number tested in the paper has been limited (mostly 2 or 3). The results would be more significant if the effectiveness of the method could be demonstrated with more complex data of larger scale.
- Please enlarge the font size in figure 3, and also adjust the opacity of the bars so that the overlapping distributions are both visible.

**Questions:**

- Should the summation in line 178 be a production instead?
- Why does the equation in line 181,
$p_{\psi}(z|\tau) = \frac{\exp \left(\prod_{(s,a) \in \tau} p_{\psi}(s,a|z)\right)}{\sum_{z'} \exp \left(\prod_{(s,a) \in \tau} p_{\psi}(s,a|z')\right)}$ hold? It would be great if the authors could elaborate more on the derivation.

**Limitations:**

The authors acknowledge two valid limitations in their work. Firstly, their MMICRL method doesn't consider potential collaboration or competition between agents to meet joint constraints. Secondly, their evaluations are conducted in virtual environments, not real-world applications, due to the absence of a suitable ICRL benchmark.

---

> ### Author Rebuttal · Authors · 2023-08-09
>
> Dear Reviewer, we sincerely value your time and effort in evaluating our work. Your insights have been valuable to our work. We have prepared comprehensive responses and clarifications to address each point you raised. We hope these responses can resolve your concerns.
>
> 1. *"The number of agent types needs to be preset, and such number tested in the paper has been limited (mostly 2 or 3). The results would be more significant if the effectiveness of the method could be demonstrated with more complex data of larger scale."*
>
> **Our Response**. We have supplemented our findings with further experimental results, enclosed in the attached PDF. This includes an increase in the number of ground-truth constraints and a study examining the model's performance when the number of these constraints and the preset agent types are mismatched. These additional results can provide clearer insight into our ICRL's performance.
>
> 2. *"Please enlarge the font size in Figure 3, and also adjust the opacity of the bars so that the overlapping distributions are both visible."*
>
> **Our response.** We appreciate your kind suggestions and we have revised the manuscript accordingly.
>
> 3. *"Should the summation in line 178 be a production instead?"*
>
> **Our response.** Thank you for your careful reading of our manuscript. It should be a production in line 178. We apologize for any confusion caused and appreciate the valuable suggestions.
>
> 4. *"Why does the equation in line 181, hold? It would be great if the authors could elaborate more on the derivation."*
>
> **Our response.** We're thankful for your suggestions. Indeed, this equation utilizes the softmax representation. We've developed an agent-conditioned density model, $p(\tau|z)=\prod_{(s,a)\in\tau} p(s,a|z)$, to calculate the density of state-action pairs. However, the agent identifier in MMICRL uses expert trajectories as input to predict the most likely agent generating these trajectories. We need to convert the density estimate to the agent identifier $p_\phi(z|\tau)$, a transformation commonly achieved via the softmax function.

---

> > ### Comment · Reviewer_KNkP · 2023-08-19
> > **Thanks for the rebuttal**
> >
> > I thank the authors for their response.
> > For Q1, the supplemented materials do not address my concern that the algorithm is only demonstrated with very limited number of agent types (fewer than 4). In realistic scenarios, this number could be substantially larger, in which case it is not clear whether the algorithm can perform well.
> > For Q4, given the likelihood $p(\tau|z)=\prod_{(s,a)\in\tau} p(s,a|z)$, the posterior $p_\phi(z|\tau)$ should be given by Bayes' Theorem, which does not involve the exponential, considering that $p(\tau|z)$ is not a logit score but already a probability. I'd suggest the authors to further clarify this in the article, or change the notation and do not call the softmax quantity the posterior $p_\phi(z|\tau)$.

---

> > > ### Author Response · Authors · 2023-08-20
> > > **Thanks for your feedback**
> > >
> > > Dear Reviewer, thanks again for taking the time to review our paper and providing valuable feedback. We greatly appreciate your important concerns and suggestions. We hope these concerns are addressable with the following points:
> > >
> > > 1. "*In realistic scenarios, this number could be substantially larger, in which case it is not clear whether the algorithm can perform well*"
> > >
> > > - **Response**. We would like to clarify that the additional experiments in the realistic environment were conducted primarily to address the concerns raised by other reviewers, who requested more real-world studies. These were not intended to respond to Q1. We apologize for any confusion our previous communication may have caused.
> > >
> > > In fact, real-world environments pose more challenges, requiring longer training times. Due to the constraint of providing a response within a week, we only considered distance constraints of 20m and 40m to validate the algorithm's applicability in autonomous driving scenarios. Nonetheless, our algorithm has the capability to handle a wider range of agent types (see "Experiments with a Larger Variety of Agent Types.").
> > >
> > > 2. "*For Q1, the supplemented materials do not address my concern that the algorithm is only demonstrated with a very limited number of agent types (fewer than 4).*"
> > >
> > > - **Response**. In the paper and the additional experiments in the attached PDF, we have investigated the performance of MMICRL in environments featuring 2, 3, and 4 distinct agent types. While we acknowledge the potential benefits of expanding this scope to encompass 5, 6, 7, and so forth, it's worth noting that the primary objective of this paper is to introduce a framework that facilitates ICRL from multiple expert types. In similar contexts, earlier research on IRL or imitation learning [10] typically involved **a maximum of 3 agent types** in their experiments.
> > >
> > > Considering that increasing the number of agent types in the demonstration dataset will increase the difficulty of "unsupervised agent identification", it is important to **explore the scalability limits** of the algorithm by increasing the number of agent types. Although we **have initiated this study and anticipate including additional results**, it is important to clarify that this investigation is not the **primary objective** of our current work.
> > >
> > > [10] Li, Yunzhu, Jiaming Song, and Stefano Ermon. "Infogail: Interpretable imitation learning from visual demonstrations." Advances in neural information processing systems 30 (2017).
> > >
> > > 3. "*For Q4, given the likelihood $p(\tau|z)=\prod_{(s,a)\in\tau} p(s,a|z)$, the posterior $p_\phi(z|\tau)$ should be given by Bayes' Theorem, which does not involve the exponential, considering that is not a logit score but already a probability.*"
> > >
> > > - **Response**. Thank you for your insightful suggestion. In fact, we can employ Bayes' Theorem in our formulation, expressed as:
> > >
> > > $$p_\phi (z|\tau) = \frac{p(z)\cdot p_\phi(\tau|z)}{\sum_z p(z)\cdot p_\phi(\tau|z)}$$
> > >
> > > where $p(z) = \frac{1}{|\mathcal{Z}|}$ denotes a uniform prior. This representation is similar to our "softmax", but without the exponential term in both numerator and denominator. It's important to note that in our formula, we leverage the probability as the logit for the softmax function. This is done to ensure consistency with the max operator we utilize in line 184, with which we construct the identified dataset.
> > >
> > > In response to your feedback, we'll enhance our paper by **incorporating the Bayesian representation**. Additionally, we will try to redefine the term 'softmax' to avoid any potential confusion.

---

> > > > ### Comment · Reviewer_KNkP · 2023-08-20
> > > >
> > > > Thanks for the authors further responses which address my concerns. I will keep my ratings.

---

> > > > > ### Author Response · Authors · 2023-08-20
> > > > > **Thanks once again**
> > > > >
> > > > > Dear Reviewer, we would like to express our gratitude for your insightful comments. We deeply appreciate the dedication and contribution you have dedicated to reviewing our paper and we will ensure that all the necessary details are thoroughly addressed and incorporated into the revised manuscript.

---

### Author Rebuttal · Authors · 2023-08-09

Dear Reviewers, Area Chairs, and Program Chairs,

We sincerely appreciate the valuable comments and suggestions provided, as they have been instrumental in enhancing our work. In light of your feedback, we can improve our work with detailed clarifications, comprehensive explanations, and supplementary experimental results. To facilitate understanding and mitigate any potential ambiguity, we provide a summary of the major updates below:

1. **Validating the Robustness with Additional Experiment.** To address the concerns raised by reviewer ZL9H, we evaluate our model's performance across different numbers of agent types. The algorithm maintains its efficiency even when the parameter surpasses the actual number of agent types. However, a parameter smaller than the actual count may influence the performance of constraint inference. In response to reviewer KNkP's concerns, we test in a Gridworld environment with a larger number of agent types, where the algorithm consistently exhibits satisfying performance. The results can be found on the attached one-page PDF.

2. **Clarifying the Advantage of MMICRL.** As it is requested by the reviewer ieEm, we clarify the distinctions between MMICRL and other methods, given that they are proposed for different tasks. Additionally, we provide a concise summary of the unique advantages offered by the MMICRL algorithm.

3. **Justifying Various Design Decisions**. We furnish the comprehensive conditions for Proposition 4.2. Additionally, we illuminate our motivation for adopting the contrastive estimate method, which effectively handles the balance between reward maximization and density maximization.

4. **Responding to Queries About Evaluation Process and Problem Formulation**. We detail our approach of allocating the identified agents to distinct expert agent types as part of the evaluation process. Besides, We introduce why explaining conflicting behaviors among expert trajectories through reward inference with IRL is a challenging task, so we think MMICRL proves to be an efficacious instrument for modeling the behaviors of multiple agents.

5. **Adding Results in Realistic Environments**. As suggested by reviewers ZL9H and Gwnr, we have added the experiment results of our MMICRL method in the realistic environment of autonomous driving on highways. We mainly study the car distance constraints (check results in the attached one-page PDF).

As far as we know, we are the first to differentiate multiple constraints corresponding to various types of agents. In order to demonstrate the generality of our method compared to other baselines, we examine the capability of MMICRL to accurately perform multi-modal constraint inference in both discrete and continuous environments. We believe that our benchmark can facilitate the development of more mature multi-model ICRL algorithms in other meaningful settings.

---

### Decision · Program_Chairs · 2023-09-21

**Decision:**

Accept (poster)

**Comment:**

The paper proposes a new approach that can deal with having differing constraints in the expert demonstrations used for Inverse Constraint Reinforcement Learning (ICRL). The approach is extensively evaluated experimentally.

The reviewer found the method interesting and sound. Initial concerns included having to specify the number of modes and only being tested on low numbers of modes, the need for more realistic experiments, some unclear details, and missing theoretical guarantees.

The authors provided additional experimental results, details, and in-depth discussion in the rebuttal. Those addressed the (major) concerns of the reviewers.

3 reviewers are now in favor of accepting the paper, the "borderline reject" reviewer did not react after the rebuttal but his/her concerns have been well addressed during the discussion phase. While the paper certainly has limitations (as also pointed out by the authors), I believe it is an important step forward and should be presented to the community, such that future research can build on top of it.